# Self-reporting photodynamic nanobody conjugate for precise and sustainable large-volume tumor treatment

Yingchao Chen[1,2], Tao Xiong[1,2], Qiang Peng[1,2], Jianjun Du[1,2], Wen Sun [1,2], Jiangli Fan [1,2] ✉ & Xiaojun Peng [1,2]

Nanobodies (Nbs), the smallest antigen-binding fragments with high stability and affinity derived from the variable domain of naturally occurring heavy-chain-only antibodies in camelids, have been shown as an efficient way to improve the specificity to tumors for photodynamic therapy (PDT). Nonetheless, the rapid clearance of Nbs in vivo restricts the accumulation and retention of the photosensitizer at the tumor site causing insufficient therapeutic outcome, especially in large-volume tumors. Herein, we develop photodynamic conjugates, MNB-Pyra Nbs, through site-specific conjugation between 7D12 Nbs and type I photosensitizer MNB-Pyra (morpholine-modified nile blue structure connected to pyrazolinone) in a 1:2 ratio. The photosensitizers with long-term retention can be released at the tumor site by reactive oxygen species cleavage after illumination, accompanied with fluorescence recovery for self-reporting the occurrence of PDT. Ultimately, a single dose of MNB-Pyra Nbs demonstrate highly effective tumor suppression with high biosafety in the large-volume tumor models after three rounds of PDT. This nanobody conjugate provides a paradigm for the design of precise long-time retention photosensitizers and is expected to promote the development of PDT.

Photodynamic therapy (PDT) has emerged as a promising cancer treatment modality due to its minimal invasiveness, repeatability, negligible drug resistance, and high spatiotemporal precision[1–3]. Photosensitizers can quickly sensitize oxygen to generate reactive oxygen species (ROS) upon light irradiation, resulting in cancer cell death accompanied by destruction of the tumor vasculature and induction of an inflammatory response for tumor ablation[4–6]. Despite the progress made in the clinical application of traditional photosensitizers such as porphyrins and phthalocyanine derivatives[7,8], photosensitizers are still plagued by the issues of off-target toxicity, tumor hypoxia, and inadequate accumulation or poor retention at the tumor site, which hamper the efficiency of PDT[9–12].

To achieve high tumor specificity, antibody–drug conjugates (ADCs) has garnered lots of attention as a targeting approach with high biosafety due to the specific interaction between antibodies and cell membrane receptors. At present, the development of ADCs is mainly based on monoclonal antibodies (mAbs) conjugates, including the antibody–photosensitizer conjugates in PDT[13,14]. For example, the conjugation of cetuximab mAb and IRdye700DX photosensitizer, has been extensively studied for the treatment of recurrent head and neck cancer[9,15]. However, the challenges associated with specific conjugation and the high production costs of mAbs have limited the development of ADCs[16–18]. In contrast, nanobodies (Nbs), the smallest antigen-binding fragments derived from heavy chain-only antibodies that are exclusively found in camelids (VHH) and cartilaginous fish (VNAR), have gained much interest[19,20]. Their single-domain nature allows their direct expression in bacterial systems, making them easier to prepare and modify as needed. Compared to conventional mAbs

---

[1]State Key Laboratory of Fine Chemicals, Frontiers Science Center for Smart Materials Oriented Chemical Engineering, Dalian University of Technology, No. 2 Linggong Road, Dalian 116024, China. [2]Liaoning Binhai Laboratory, Dalian 116023, China. ✉e-mail: fanjl@dlut.edu.cn

with a size of approximately 150 kDa, the small size of Nbs (approximately 15 kDa) enables them to reach less accessible antigens while maintaining high affinity and stability[21,22]. Sabrina Oliveira and her colleagues first published on the approach of nanobody-targeted PDT in 2014[23], and some other nanobody-targeted PDT related works were reported later[24–27]. Nevertheless, Nbs are rapidly removed from mammals via the circulation with an elimination half-life of 60–90 min due to their small size[17,28,29].

In terms of the sustainability and phototoxic side effects of treatment, the cycling and clearance of photosensitizers in PDT must be considered. Excessively fast clearance of the photosensitizer results in insufficient drug accumulation, hindering subsequent sustained treatment, while slow clearance leads to a long drug clearance period, potentially causing severe phototoxic side effects[30–32]. In this regard, accurately delivering long-term resident photosensitizers to the tumor site via Nbs may be an ideal strategy to reduce drug enrichment in normal tissues, enhance drug retention in tumor tissues, and enable precise and sustainable PDT, especially for the treatment of large-volume tumors.

Epidermal growth factor receptor (EGFR) is a heavily glycosylated transmembrane receptor tyrosine kinase that is closely associated with tumorigenesis and cancer proliferation[33,34]. Due to its overexpression in tumor cells, EGFR is an important target in cancer therapy[35–37]. The nanobody 7D12 specifically binds to the ligand-binding domain III of the EGFR and is subsequently endocytosed into the cells[18,38,39]. The benzophenothiazine structure of morpholine-modified nile blue (MNB) has been classified as a type I photosensitizer, which efficiently produces superoxide radicals ($O_2^{-\cdot}$) by electron transfer after photoexcitation[40,41]. Then, the MNB-Pyra molecule was constructed to covalently modify the Nbs by connecting MNB with the pyrazolone (Pyra) structure through a ROS-cleavable linker[42,43]. The good stability and solubility of MNB-Pyra in aqueous solution indicate that this molecule has high protein compatibility, which is conducive to bioconjugation.

In this work, the MNB-Pyra Nbs conjugate is constructed by connecting the type I photosensitizer MNB-Pyra with 7D12-fGly via a Knoevenagel−Michael tandem reaction, as shown in Fig. 1, in which 7D12-fGly is a modified form of 7D12 with a C-terminus aldehyde handle for specific and quantitative conjugation. MNB-Pyra Nbs exhibit significant fluorescence quenching due to the π − π stacking interactions of the planar molecule MNB-Pyra after site-specific modification of the Nbs with MNB-Pyra in a 1:2 ratio. Upon light irradiation, fluorescence recovery can be observed since the monomer photosensitizer is released due to cleavage by ROS, displaying the self-reporting ability of MNB-Pyra Nbs during PDT. Moreover, the MNB-Pyra Nbs demonstrate efficient targeting ability and PDT efficiency in vitro and in vivo. Interestingly, MNB-Pyra Nbs are almost completely cleared 24 h after intravenous injection without illumination, but the long-term retention characteristics of the benzophenothiazine compound ensures that the released monomer photosensitizer remained precisely at the tumor site for 5 days after light irradiation. Significantly, MNB-Pyra Nbs efficiently inhibit tumor growth in the large-volume tumor model after single-dose administration and continuous PDT. Consequently, this photodynamic nanobody conjugate provides a paradigm for the development of precise long-term resident photosensitizer.

## Results

### The photophysical properties and ROS generation capacity of dimerized MNB-Pyra

The synthesis route of MNB-Pyra was shown in Supplementary Fig. 1. The MNB-Pyra dimer was synthesized by leveraging the reactivity of the pyrazolone to aldehydes, in which MNB-Pyra dimerized upon reaction with isobutyraldehyde via a Knoevenagel−Michael tandem reaction (Fig. 2a). All intermediates and products were characterized by mass spectrometry (MS) and nuclear magnetic resonance (NMR)

spectroscopy in Supplementary Figs. 33–42. As shown in Fig. 2b, c, MNB-Pyra displayed a main absorption band at 645 nm and strong fluorescence emission at 708 nm. In comparison, the MNB-Pyra dimer exhibited a blueshifted band at 600 nm and significant fluorescence quenching, with its absolute fluorescence quantum yield ($\Phi_f$) decreasing from 4.6% to 0.9% (Supplementary Table 1).

To explore the fluorescence quenching mechanism and gain deeper insights into the structural arrangement of the MNB-Pyra dimer, we performed a conformational search of the MNB-Pyra dimer in water using the GFNn-xTB method for molecular dynamics calculations and geometry optimization at the B97-3c density functional approximation level[44,45]. The lowest energy conformation of the MNB-Pyra dimer in water (99.8% abundance) presented with a π-stacked geometrical arrangement and orbital overlap with a center-to-center distance of 4.4 Å and a flip angle of θ = 55.2° (Fig. 2d, e). The noncovalent intramolecular interaction was analyzed using an independent gradient model, which also revealed strong interchromophoric π − π interactions (Fig. 2e). These results confirm that the MNB-Pyra dimer is a self-quenching foldamer with a dimerization-caused quenching (DCQ) mechanism. The foldamer structures of most DCQ fluorophores will open after solubilization in organic solvents, which occurs along with the abolishment of aggregation and recovery of fluorescence emission. However, the MNB-Pyra dimer maintained DCQ effects with weak fluorescence emission even in polar solvents (e.g., acetonitrile and methanol), highlighting the strong H-bonding interactions between the MNB-Pyra molecules in the dimer (Supplementary Fig. 2). In addition, the fluorescence quenching of the MNB-Pyra dimer remained unperturbed in biological media in the presence of serum proteins (Supplementary Fig. 3), indicating the stability of the self-quenching MNB-Pyra dimer structure in a biological context.

Next, the ROS generating ability of the molecules was evaluated with fluorescent probes. 1,3-Diphenylisobenzofuran (DPBF) and 9,10-Anthracenediyl-bis(methylene) dimalonic acid (ABDA) were employed to detect the generation of $^1O_2$, where dihydroethidium (DHE) and dihydrorhodamine 123 (DHR 123) served as indicators of $O_2^{-\cdot}$ with fluorescence emission at 600 nm and 526 nm. As anticipated, both the MNB-Pyra and MNB-Pyra dimers exhibited negligible $^1O_2$ generation (Fig. 2f and Supplementary Fig. 4) and efficient $O_2^{-\cdot}$ generation (Fig. 2g and Supplementary Fig. 5), affirming their efficiency as type I photosensitizers. In addition, the generation of $O_2^{-\cdot}$ by MNB-Pyra was confirmed with the indicator 5,5-dimethyl-1-pyrroline-N-oxide (DMPO) by electron paramagnetic resonance (EPR) spectroscopy, and a characteristic paramagnetic adduct of $O_2^{-\cdot}$ was observed (Fig. 2h). The high photostability of MNB-Pyra was validated by the absorption with only a slight decrease after the light irradiation (630 nm, 50 mW/cm²) for 60 min, and then its ROS generation ability was also not affected (Supplementary Fig. 6). Subsequently, a redshifted absorption band and significant fluorescence enhancement of the MNB-Pyra dimer were observed after illumination, as shown in Fig. 2I, j, indicating depolymerization of the dimer and disruption of the DCQ effect. The cleaved products were further analyzed by high resolution mass spectrometry (HRMS), which demonstrated that the MNB-Pyra dimer was consumed and that the monomer photosensitizer (m/z = 479.1568 for [M]⁺) was released (Supplementary Fig. 7). As a result, the MNB-Pyra dimer can self-report the occurrence of PDT by the fluorescence enhancement after the release of monomer photosensitizers.

### Preparation of the Nbs conjugate via in situ MNB-Pyra dimerization

Instead of presynthesizing MNB-Pyra dimers for bioconjugation, the MNB-Pyra molecule was dimerized in situ at the C-terminus of 7D12-fGly utilizing the reactivity of the pyrazolone moiety to the aldehyde tag, which was installed using a formylglycine-generating enzyme (FGE)[46,47]. MNB-Pyra Nbs were formed by reacting 7D12-fGly with MNB-Pyra in 2-morpholinoethanesulfonic acid buffer (MES buffer, pH 6.5) at

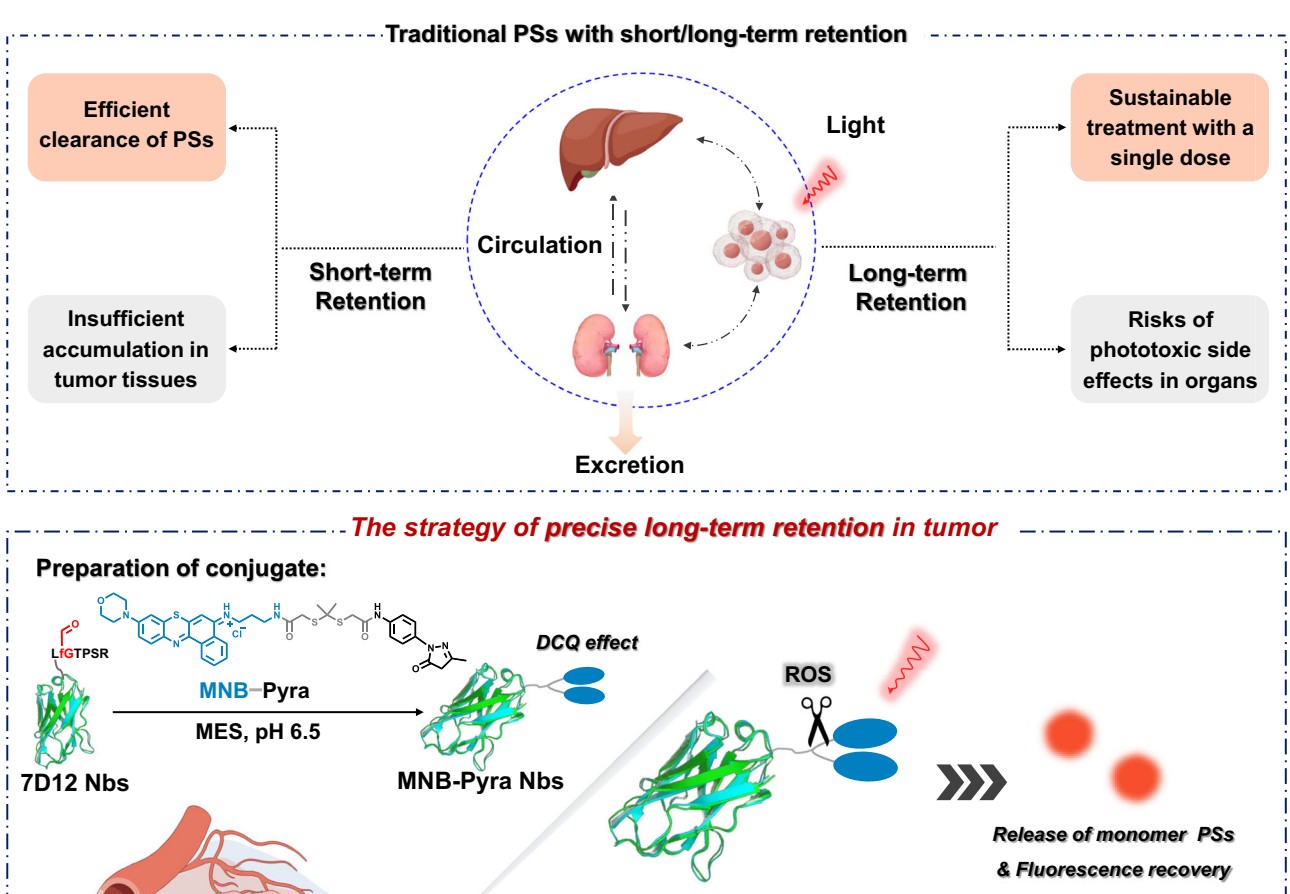

**Fig. 1 | Mechanism of MNB-Pyra Nbs in vivo.** Construction of the self-reporting photodynamic nanobody conjugate to realize self-reporting and precise long-term retention in tumor tissue for sustainable PDT treatments. Schematic created with Figdraw (ID: IIUTAd4f1d) and PowerPoint.

room temperature (Fig. 3a). Sodium dodecyl sulfate–polyacrylamide gel electrophoresis (SDS–PAGE) and high-performance liquid chromatography–high resolution mass spectrometry (HPLC−HRMS) were used to characterize the synthesized MNB-Pyra Nbs. As displayed in Fig. 3b, the conjugate displayed a weak fluorescence band with no significant band migration compared to the original 7D12-fGly protein. HPLC−HRMS analysis of the protein (Fig. 3c) also confirmed the formation of MNB-Pyra Nbs (mass of 7D12-fGly: 17990.770; mass of MNB-Pyra Nbs: 19536.219), in which each 7D12-fGly nanobody was conjugated with two MNB-Pyra molecules with a conjugation efficiency exceeding 95% in 24 h.

MNB-Pyra Nbs exhibited good water solubility, which the high concentration of MNB-Pyra Nbs (0.5 mM, 100 μL) in aqueous solution remained clear and transparent without solid precipitation after being placed at 4 °C for 5 days (Supplementary Fig. 8). The dynamic light scattering (DLS) analysis showed that the hydrodynamic diameter of 7D12-fGly Nbs and MNB-Pyra Nbs was 5.3 nm and 6.1 nm respectively (Fig. 3d), suggesting the particle size of the nanobody did

not change significantly after the conjugation. Besides, the negligible fluorescence changes of the conjugate after the co-incubation with enzymes/other biomolecules reflected its high stability under physiological conditions (Supplementary Fig. 9). The binding affinity of 7D12-fGly Nbs and MNB-Pyra Nbs to EGFR were evaluated by biolayer interferometry (BLI), which analyzes biomolecular interactions via an optical biosensing technology[48–50]. As shown in Supplementary Fig. 10, 7D12-fGly Nbs and MNB-Pyra Nbs showed comparable EGFR binding affinity, which have nanomolar range of dissociation constant ($K_D$) value of $1.79 \pm 0.27$ nM and $24.33 \pm 0.65$ nM, respectively. Though the introduction of photosensitizers slightly decreased the binding affinity of Nbs, the conjugate still exhibited significant EGFR binding capacity in the cell-based ELISA analysis (Supplementary Fig. 11). Therefore, it is convinced that the Nbs can reserve its high binding affinity after the site-specific conjugation, achieving the delivery of photosensitizers toward EGFR overexpressed cells.

Moreover, MNB-Pyra dimerization in the nanobody conjugate was expected to realize fluorescence quenching like MNB-Pyra dimer.

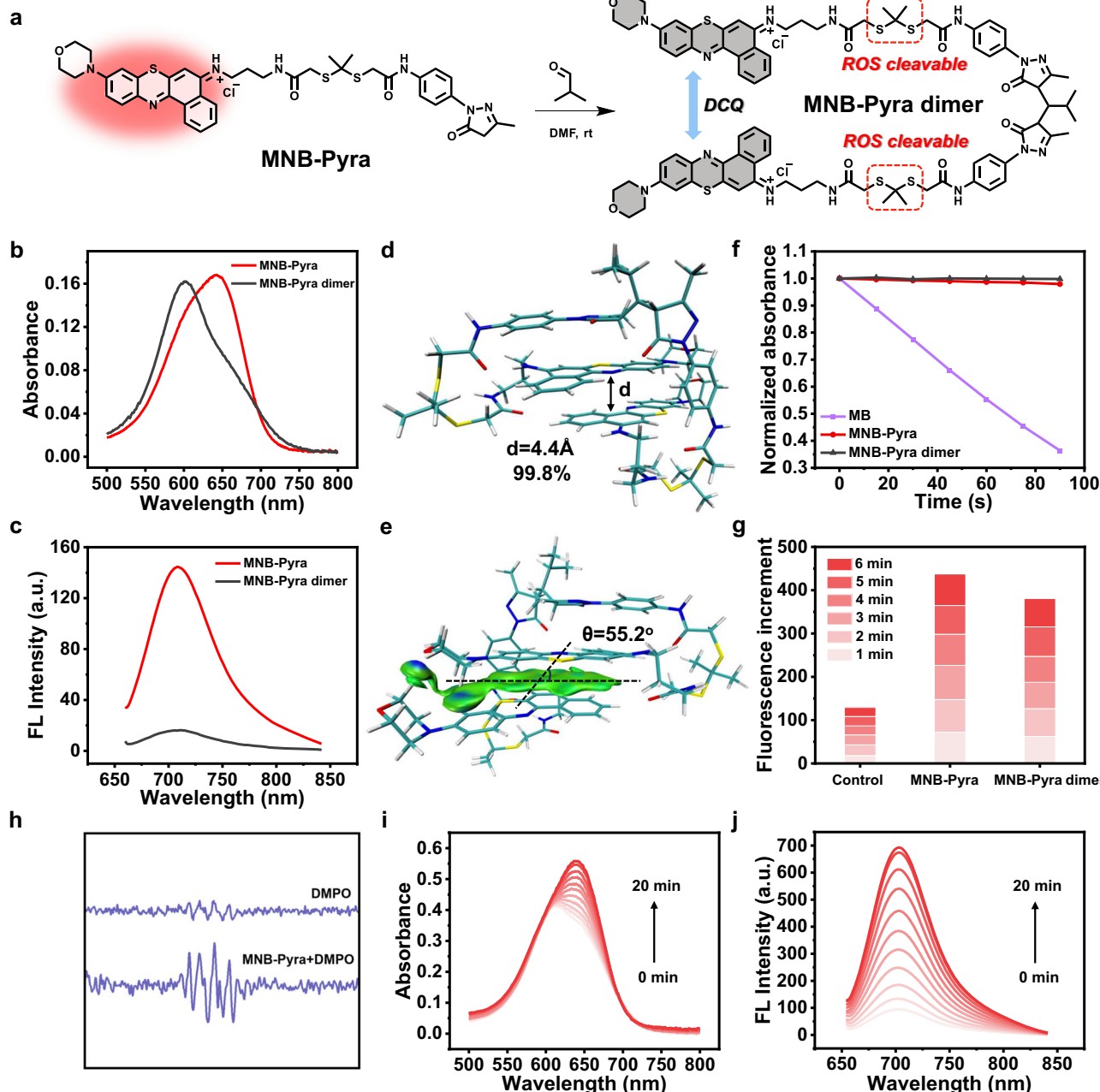

**Fig. 2 | Spectral analysis of MNB-Pyra and MNB-Pyra dimer. a** Schematic representation of MNB-Pyra dimer synthesis. **b**, **c** Absorbance and fluorescence spectra of MNB-Pyra (5 μM) and MNB-Pyra dimer (5 μM) in PBS solution. **d** Calculated conformation of MNB-Pyra dimer in water. **e** Analysis of the intramolecular weak interactions within MNB-Pyra dimer. The π − π interaction is denoted as the green disk. **f** DPBF attenuation curve of MNB-Pyra (5 μM) and MNB-Pyra dimer (5 μM) compared with MB (5 μM). **g** Generation ability of $O_2^{-\bullet}$ using DHR123 as a fluorescence probe. **h** EPR test with DMPO for $O_2^{-\bullet}$ characterization. **i–j** Absorbance and fluorescence recovery of the MNB-Pyra dimer (10 μM) after 630 nm light irradiation (30 mW/cm²). Source data are provided as a Source Data file.

Thus, the absorption and fluorescence spectra of the MNB-Pyra Nbs were investigated in phosphate-buffered saline (PBS, pH 7.4). As shown in Fig. 3e, the maximum absorption peak of MNB-Pyra Nbs exhibited a noticeable blueshift from 645 nm to 600 nm in comparison to the spectrum in MNB-Pyra. Significant fluorescence quenching was observed in Fig. 3f. The absolute fluorescence quantum yield of MNB-Pyra Nbs (0.8%) in aqueous solution closely resembled that of the MNB-Pyra Nbs dimer ($\Phi_f = 0.9\%$, Supplementary Table 1), confirming the significant reduction in fluorescence emission. The dimer-like spectral features of the MNB-Pyra Nbs verified the aggregation of MNB-Pyra within the nanobody conjugates. What's more, the increased single peak of blue-shifted absorption bands was observed in the

absorbance spectra of MNB-Pyra dimer and MNB-Pyra Nbs with increasing concentration (Supplementary Fig. 12), where the aggregation degree is independent of their concentration. This result declared the aggregation mechanism of DCQ in MNB-Pyra Nbs. Similarly, MNB-Pyra Nbs were able to recover their fluorescence emission upon exposure to 630 nm light (Fig. 3g), indicating that the MNB-Pyra dimer within the conjugate had depolymerized. The cleaved products were identified by HRMS analysis, and the product (m/z = 479.1566 for [M]⁺) was the same monomer photosensitizer as that detected after MNB-Pyra dimer cleavage (Supplementary Fig. 13). The release rate of MNB-Pyra Nbs after illumination was investigated by the absorption spectrum in Supplementary Fig. 14, which exhibited a high release

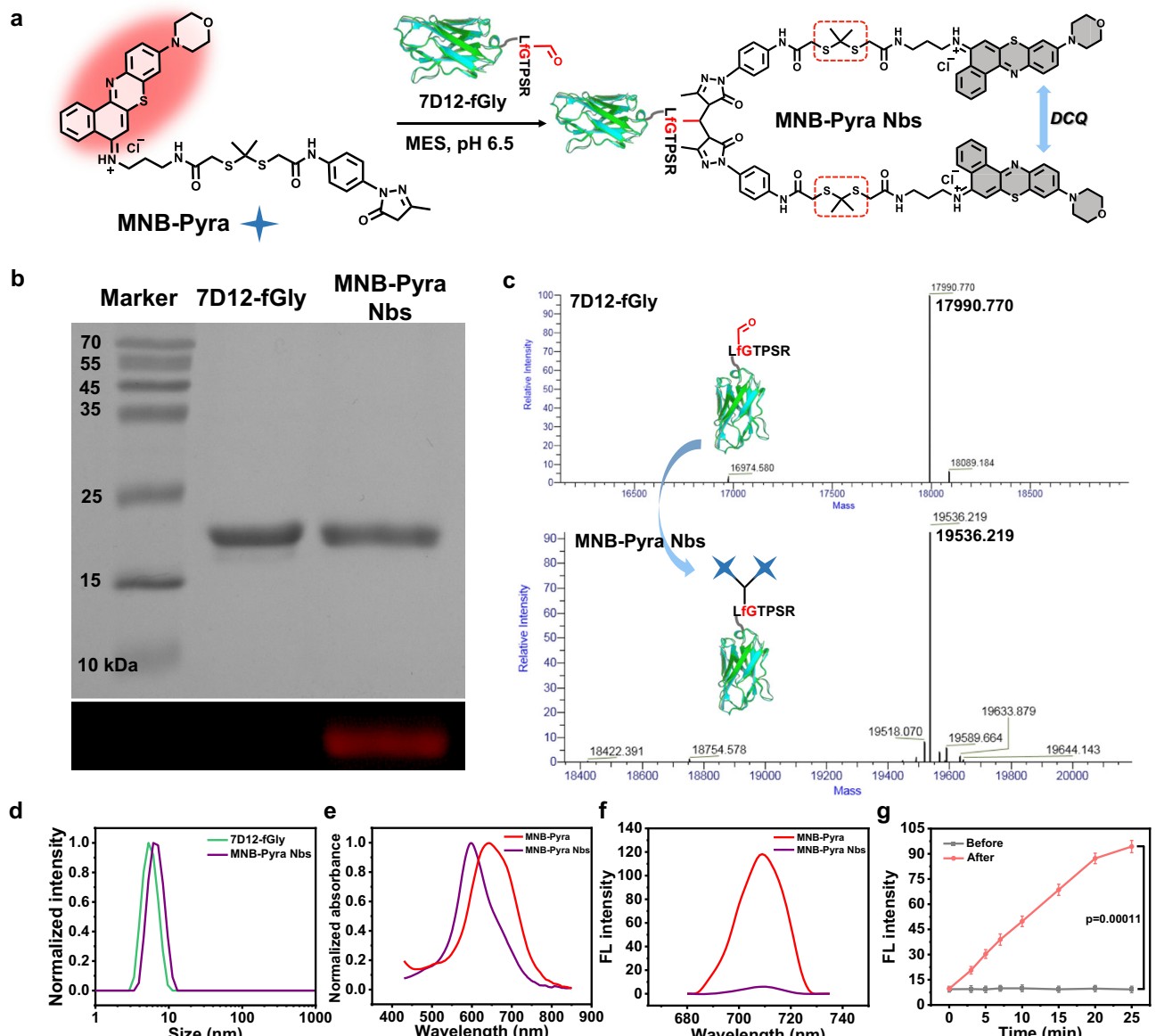

**Fig. 3 | The characterization of MNB-Pyra Nbs and its spectral analysis.**
**a** Schematic representation of the preparation of MNB-Pyra Nbs. **b** SDS-PAGE analysis of the protein conjugate MNB-Pyra Nbs. **c** Protein mass spectrometry analysis of MNB-Pyra Nbs. **d** Hydrodynamic size of 7D12-fGly and MNB-Pyra Nbs by dynamic light scattering. **e**, **f** Normalized absorbance and fluorescence spectra of MNB-Pyra (10 μM) and MNB-Pyra Nbs (5 μM). **g** Fluorescence recovery of MNB-Pyra Nbs after 630 nm light irradiation (30 mW/cm²), $n = 3$ experimental replicates, data are shown as mean ± SD. Statistical significance in (**g**) was calculated via one tailed Student's t test. $P < 0.05$ is considered to be statistically significant. Source data are provided as a Source Data file.

efficiency about 83%. The successful fluorescence recovery of the MNB-Pyra Nbs suggested their efficient ROS generation ability under light irradiation. Consequently, this photodynamic nanobody conjugate can self-report the production of ROS and the release of monomer photosensitizers during PDT through the increasing fluorescence intensity.

**Specificity of MNB-Pyra Nbs for EGFR-positive cells**
We reasoned that attaching MNB-Pyra to an anti-EGFR nanobody (7D12-fGly) would endow the molecule with high selectivity for EGFR-overexpressing cells. Therefore, four different cell lines with different EGFR expression levels were employed for cell experiments, including human epidermoid carcinoma (A431), human cervical carcinoma (HeLa), human lung cancer (A549) and mouse embryonic cells (NIH-3T3). Western blot assays confirmed that A431 and HeLa cells are EGFR-positive cells, in which A431 exhibited high

EGFR expression and HeLa had moderate expression, while A549 and NIH-3T3 cells had almost negligible EGFR expression (Fig. 4a). First, the cytocompatibility of the MNB-Pyra Nbs was investigated. More than 90% of each type of cell survived after incubation with increasing concentrations of MNB-Pyra Nbs for 24 h without light irradiation (Fig. 4b), manifesting the satisfied biocompatibility of the conjugate in vitro. Next, the uptake efficiency of MNB-Pyra Nbs by different cells was evaluated using confocal laser scanning microscopy (CLSM) and flow cytometry. EGFR-positive A431 and HeLa cells exhibited significant cellular uptake, with increasing fluorescence signals in the cells over time (Fig. 4c and Supplementary Fig. 15). In contrast, negligible fluorescence signals were observed in A549 and NIH-3T3 cells, indicating minimal cellular uptake of MNB-Pyra Nbs. Flow cytometry quantification of MNB-Pyra Nbs (0.2 μM) uptake further supported the significantly higher fluorescence signals in A431 and HeLa cells (Fig. 4d). These results confirmed the

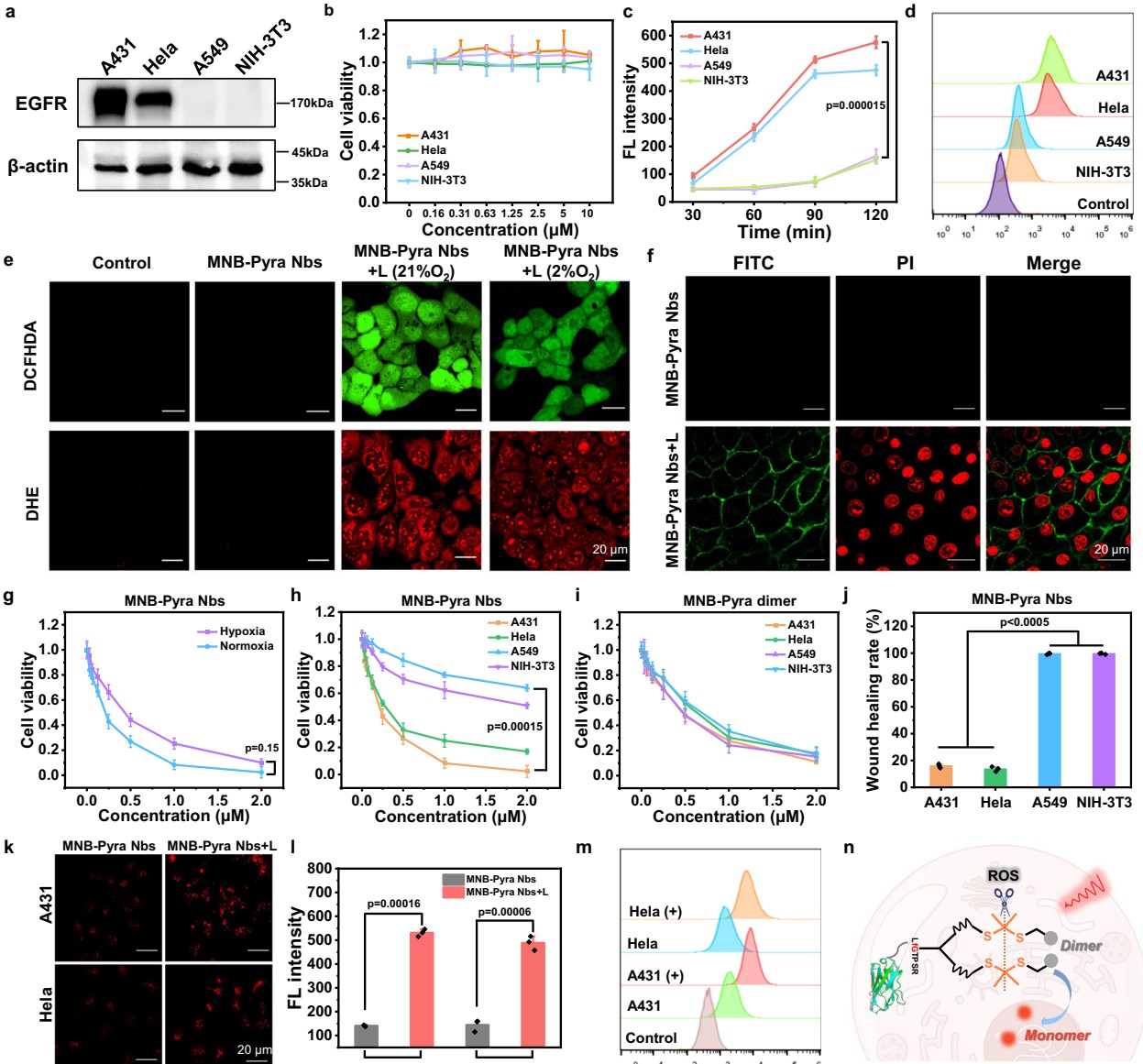

**Fig. 4 | Targeted imaging, self-reporting, and PDT effect of MNB-Pyra Nbs in cells. a** Western blot analysis of the EGFR expression in A431, HeLa, A549, and NIH-3T3 cells. **b** Cell viability of different cells incubated with MNB-Pyra Nbs at various concentrations in dark ($n$ = 4 experimental replicates). **c** Quantification of the fluorescence intensity of MNB-Pyra Nbs in different cells ($n$ = 3 experimental replicates). **d** Flow cytometry analysis of MNB-Pyra Nbs uptake in cells. **e** ROS generating detection of MNB-Pyra Nbs in A431 cells under normoxia and hypoxia conditions with DCFH-DA ($E_x$: 488 nm, $E_m$: 500−550 nm) and DHE ($E_x$: 560 nm, $E_m$: 600−630 nm). **f** Apoptosis imaging of A431 cells using Annexin V-FITC ($E_x$: 488 nm, $E_m$: 490−530 nm) and PI ($E_x$: 488 nm, $E_m$: 600−700 nm) kit. **g** Cell viability of A431 cells after PDT with MNB-Pyra Nbs under normoxia or hypoxia conditions ($n$ = 4 experimental replicates). **h**−**i** MTT assays of MNB-Pyra Nbs and MNB-Pyra dimer in different cells after PDT ($n$ = 4 experimental replicates). **j** Wound healing rate of different cells after PDT with MNB-Pyra Nbs ($n$ = 3 experimental replicates). **k** Fluorescence imaging of MNB-Pyra Nbs (0.2 μM) in A431 and HeLa cells with or without light irradiation. **l** Quantification of the fluorescence intensity in A431 and HeLa cells in figure ($n$ = 3 experimental replicates). **m** Flow cytometry analysis of fluorescence intensity in cells with or without light irradiation after incubated with MNB-Pyra Nbs (0.2 μM) for 2 h. **n** Schematic representation of MNB-Pyra Nbs cleaved by ROS in the cells. Schematic diagram (**n**) created with Figdraw (ID: IIU-TAd4f1d) and PowerPoint. Data are shown as mean ± SD (**b, c, g, h, i, j, l**). Statistical significance in (**c, g, h, j, l**) was calculated via one tailed Student's $t$ test. $P$ < 0.05 is considered to be statistically significant. Scale bar = 20 μm. **d, e, f, k, m** Each experiment was repeated three times with similar results. Source data are provided as a Source Data file.

high selectivity of MNB-Pyra Nbs for EGFR-overexpressing cells. Notably, neither the MNB-Pyra molecule nor the MNB-Pyra dimer exhibited any cell selectivity, as there were no observable differences in cellular fluorescence after various cell types were treated with these molecules (Supplementary Fig. 16). Therefore, modifying MNB-Pyra with the 7D12-fGly nanobody enables MNB-Pyra Nbs to specifically target EGFR-overexpressing cancer cells, thereby minimizing potential side effects during cancer therapy.

Moreover, the intracellular localization of MNB-Pyra Nbs was investigated using commercial dyes. As shown in Supplementary Fig. 17a, the intracellular fluorescence of MNB-Pyra Nbs overlapped well with the green fluorescence of LysoTracker Green with a colocalization coefficient (Pearson's correlation) of 0.871. Furthermore, the released photosensitizers after illumination were also located in the lysosomes and could product ROS efficiently inside cells with re-illumination (Supplementary Figs. 17b, c).

### Self-reporting ability and PDT effect of MNB-Pyra Nbs in vitro

ROS production by the MNB-Pyra Nbs in cells after illumination was confirmed with fluorescent probes, including the reactive oxygen indicator 2,7-dichlorodihydrofluorescein diacetate (DCFH-DA) and $O_2^{-\cdot}$ indicator DHE. As shown in Fig. 4e and Supplementary Fig. 18, bright fluorescence signals from the probes in cells were collected under both normoxia and hypoxia conditions, demonstrating efficient ROS generation. Moreover, obvious DHE fluorescence, indicating the presence of $O_2^{-\cdot}$, was also detected in the cells, implying that MNB-Pyra Nbs might have satisfactory PDT activity under hypoxia.

Subsequently, the PDT effect of MNB-Pyra Nbs was investigated. As presented in Fig. 4g, MNB-Pyra Nbs exhibited potent phototoxicity toward A431 cells under normoxia conditions with a half-maximal inhibitory concentration ($IC_{50}$) of 0.2 μM. Furthermore, cell viability was evaluated by simulating the hypoxia microenvironment of tumors (2% $O_2$). Although hypoxia conditions slightly overwhelmed the PDT outcome, treatment with MNB-Pyra Nbs still successfully led to severe cell disruption with an $IC_{50}$ of 0.4 μM (Fig. 4g), confirming their efficient PDT effect under hypoxia conditions. Additionally, a calcein AM-PI kit was used for live/dead cell imaging, where calcein AM labels live cells with green fluorescence and PI labels dead cells with red fluorescence[51]. A431 cells presented green fluorescence only after incubation with MNB-Pyra Nbs without light irradiation, suggesting the biosafety of the MNB-Pyra Nbs (Supplementary Fig. 19). In contrast, strong red fluorescence from PI was observed in cells under both normoxia and hypoxia conditions, validating the effective cell killing capability of MNB-Pyra Nbs after PDT.

Notably, MNB-Pyra Nbs had high selectivity for EGFR-positive cells during PDT. As presented in Fig. 4h, MNB-Pyra Nbs showed potent effects on A431 and HeLa cell killing with $IC_{50}$ values of 0.2 and 0.3 μM, respectively. However, the cell killing efficiency of MNB-Pyra Nbs in A549 and NIH-3T3 cells was negligible due to the low expression of EGFR in these cells. Live/dead cell images of different cell lines were also collected after PDT or without PDT. As shown in Supplementary Fig. 20, all nonilluminated cells exhibited green fluorescence only. Moreover, strong red fluorescence from PI was observed in A431 and HeLa cells after PDT, whereas A549 and NIH-3T3 cells retained bright green fluorescence. These results confirmed the high specificity of MNB-Pyra Nbs for EGFR-overexpressing cells. In comparison, the differences in phototoxicity of the MNB-Pyra dimer was negligible among the different cell lines (Fig. 4i), suggesting that the dimer does not have cell specificity.

In addition, a wound healing assay was conducted with MNB-Pyra Nbs to visualize the suppression of cell proliferation after PDT. MNB-Pyra Nbs (0.5 μM) were added when the cells in the dishes reached a density of approximately 80% and the cells were then exposed to light irradiation (630 nm, 30 mW/cm², 20 min) after incubation. Bright field images of different cells were captured at 0 h and 24 h after treatment. It was found that the proliferation of A431 and HeLa cells was significantly inhibited, with wound healing rates of 17% and 13%, respectively, while the wounds in A549 and NIH-3T3 cells healed completely (Fig. 4j and Supplementary Fig. 21). This further confirmed the specificity of MNB-Pyra Nbs to inhibit the proliferation and invasion of EGFR-overexpressing cells.

Besides, the self-reporting capability of MNB-Pyra Nbs during PDT was observed in cells. A431 and HeLa cells were exposed to 630 nm light irradiation (30 mW/cm², 10 min) after incubation with 0.2 μM MNB-Pyra Nbs for 2 h, and significantly enhanced fluorescence signals were detected in these cells (Fig. 4k, l). Additionally, flow cytometry was used to quantify the intracellular fluorescence, which was notably increased after light irradiation (Fig. 4m). Thus, it was suggested that the MNB-Pyra Nbs were cleaved in the cells after illumination, leading to the release of the monomer photosensitizer accompanied with fluorescence increasement (Fig. 4n). Meanwhile, the enhanced fluorescence signal also feedbacks the generation of ROS and indicates the occurrence of PDT.

The mechanism of cell death was investigated using an Annexin V-FITC/PI kit. During apoptosis, Annexin V-FITC specifically binds to phosphatidylserine on the outer surface of apoptotic cells, while PI enters the dead cells and emits red fluorescence upon binding to DNA[52]. Cells were incubated with the Annexin V-FITC/PI dyes for 30 min after various treatments. The control groups treated with MNB-Pyra Nbs only exhibited no fluorescence signals from either FITC or PI. However, bright green fluorescence from Annexin V-FITC and red fluorescence from PI were observed in the PDT group (Fig. 4f and Supplementary Fig. 22), indicating that MNB-Pyra Nbs induced apoptosis after light irradiation (630 nm, 30 mW/cm², 20 min). The apoptosis process was also identified with Annexin V-FITC/PI using flow cytometry (Supplementary Fig. 23). The cells in the PDT group were mainly found in the third quadrant, which represents apoptosis[53].

### Targeting and precise long-time retention of MNB-Pyra Nbs in vivo

Encouraged by the impressive specificity of MNB-Pyra Nbs for EGFR-overexpressing cells, the targeting ability of the conjugate was evaluated in tumor-bearing nude mice. An A431 tumor-bearing mouse model was constructed by subcutaneous injection of A431 cells into BALB/c nude female mice (5–6 weeks) purchased from Beijing Vital River Laboratory Animal Technology Co., Ltd. As shown in Fig. 5a and Supplementary Fig. 24a, there was no discernible fluorescence signal at the tumor site over time following intravenous injection of the MNB-Pyra dimer (0.3 mM, 100 μL). In stark contrast, mice injected with MNB-Pyra Nbs (0.3 mM, 100 μL) via the tail vein displayed a progressive increase in fluorescence at the tumor site, reaching its peak at 4 h (Fig. 5b and Supplementary Fig. 24b). Quantitative analysis corroborated the efficient tumor targeting of the MNB-Pyra Nbs in vivo (Fig. 5c). This result clearly indicated that conjugating the Nbs with the MNB-Pyra endowed the molecule with efficient targeting capability in vivo. Upon dissection at 24 h postinjection, the MNB-Pyra dimer was still mainly distributed in the liver, while the fluorescence of the MNB-Pyra Nbs in the organs was exceedingly weak, revealing the effective clearance of the conjugate. Additionally, a bilateral tumor model was constructed, and MNB-Pyra Nbs were intravenously injected (0.3 mM, 200 μL). As shown in Fig. 5d, obvious fluorescence signals from the MNB-Pyra Nbs were collected at the tumor sites after 4 h. Then, the right tumor was exposed to 630 nm light irradiation (30 mW/cm², 20 min), and the left tumor without any treatment was used as a control. As a result, a significant increase in fluorescence intensity was observed at the right tumor site (Fig. 5d, e), which indicated that the MNB-Pyra Nbs could self-report monomer release and the occurrence of PDT by the increase in fluorescence signal intensity.

Long-term retention of the benzophenothiazine compound in vivo was reported in our previous work, in which the fluorescence of photosensitizers persisted at the tumor site over 120 h[54]. Herein, the similarly long-term retention of the MNB-Pyra molecule (0.3 mM, 100 μL) in vivo was demonstrated as shown in Supplementary Fig. 25, where the fluorescence at the tumor site was retained for a week without attenuation and never spread to other sites after intratumoral injection. In fact, the prolonged retention of the photosensitizers suggests that it is difficult to clear from the body, which brings about potential phototoxicity to organs during treatment and the risk of side effects caused by the overaccumulation of drugs. Nevertheless, the clearance of MNB-Pyra Nbs within 24 h without illumination (Fig. 5b) suggested that the clearance of MNB-Pyra was improved after its conjugation with 7D12-fGly Nbs. As a proof of concept, the tumor site was exposed to 630 nm light irradiation (30 mW/cm², 20 min) after the effective enrichment of MNB-Pyra Nbs (0.3 mM, 100 μL) administered via tail vein injection (Fig. 5f). In this case, significant fluorescence enhancement was observed at the tumor site, and the fluorescence

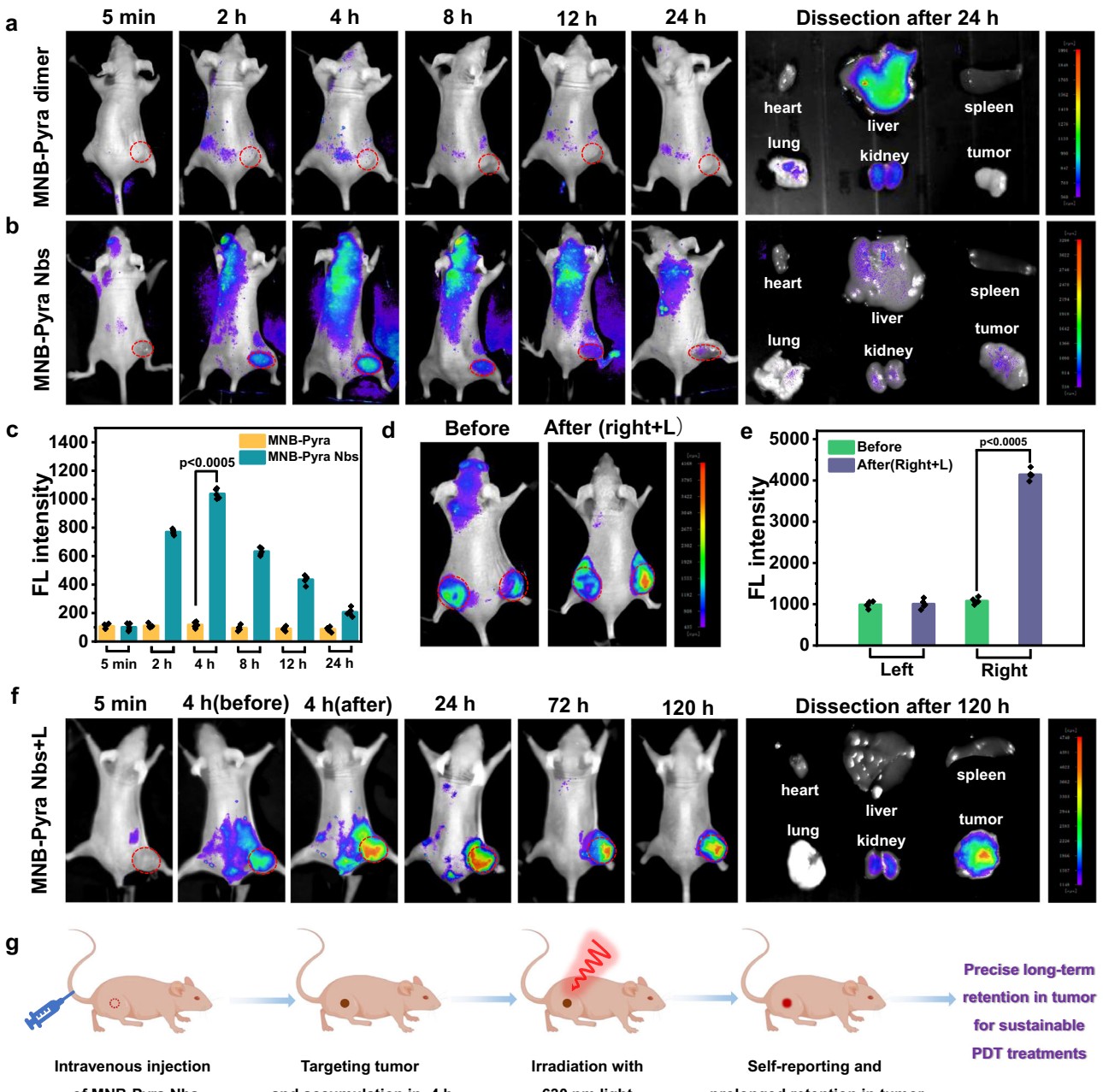

**Fig. 5 | The targeting and precise long-time retention imaging of MNB-Pyra Nbs.**
**a, b** Real-time tracking of MNB-Pyra dimer and MNB-Pyra Nbs in vivo postinjection within 24 h. **c** Fluorescence intensity of tumor site over time after the intravenous injection of MNB-Pyra dimer or MNB-Pyra Nbs ($n = 5$ mice per group). **d** Fluorescence increasement of MNB-Pyra Nbs after light irradiation (630 nm, 30 mW/cm$^2$, 20 min) in the bilateral tumor model. **e** Fluorescence intensity of tumor sites in the bilateral tumor model with or without irradiation ($n = 5$ mice per group). **f** Real-time tracking of MNB-Pyra Nbs with irradiation (630 nm, 30 mW/cm$^2$, 20 min) in vivo postinjection within several days. **g** Schematic representation of fluorescence imaging process of MNB-Pyra Nbs in vivo. Schematic created with Figdraw (ID: IIUTAd4f1d) and PowerPoint. Data are shown as mean ± SD (**c, e**). Statistical significance in (**c, e**) was calculated via one tailed Student's $t$ test. $P < 0.05$ is considered to be statistically significant. Source data are provided as a Source Data file.

signal persisted in the tumor for 5 days (Fig. 5g). Dissection analysis performed after 120 h revealed a concentrated fluorescence signal in the tumor tissue and negligible fluorescence in other organs, demonstrating that the released monomer photosensitizer achieved precise long-term retention at the tumor site after illumination.

The released monomer photosensitizers at the tumor site would be cleared in about 12 days (Supplementary Fig. 26). This drug clearance at this tumor site is different from the systemic circulatory clearance of long-term resident drugs because its drug concentrations are much lower and there is little risk of side effects through the kidney way only. Therefore, this strategy of precise long-term retention via

Nbs provides a more convenient and safer way to reduce the drug enrichment in normal tissues but enhance drug retention in tumor tissue for the achievement of sustainable PDT after a single dose of administration. Notably, the level of ROS in cancer cells is much higher than normal cells due to their increased metabolism and mitochondrial dysfunction[55,56]. In order to confirm whether the level of ROS in cancer cells will cleave the thioketal linkage, MNB-Pyra dimer (0.3 mM, 100 μL) was intratumorally injected in an A431 tumor-bearing mouse model for the long time of observation. As shown in Supplementary Fig. 27, the fluorescence signal at the tumor sites increased over time in 36 h, indicating the cleavage of the connecting chain and release of

photosensitizers. But the effective response of the thioketal linkage to ROS in 431 cells was more than 12 h, which suggested the cleavage by intracellular ROS would have negligible influence in this work.

## MNB-Pyra Nbs for large-volume tumor treatment in vivo

Single-dose PDT can generally achieve effective small tumor inhibition, whereas multiple treatments are needed for the suppression of large-volume tumors. Photosensitizers with long-term retention characteristics are suitable for the treatment of large-volume tumors, as they enable sustainable treatment after injection of a single dose, reducing the pain and inconvenience of multiple dosing. However, conventional photosensitizers with long-term retention also accumulate and are retained in organs, which brings about the risk of phototoxic side effects during subsequent continuous PDT. Therefore, the precise release of photosensitizers with long-term retention at the tumor site, along with improved photosensitizer clearance in organs, is essential to achieve sustainable PDT and minimize side effects during the ablation of large-volume tumors.

Motivated by the efficient tumor targeting and precise long-term retention of MNB-Pyra Nbs, the therapeutic effect of MNB-Pyra Nbs was investigated in mice bearing large tumors (initial volume $\approx$350 mm$^3$), as depicted in Fig. 6a. Mice in different groups treated with only PBS, light, or MNB-Pyra Nbs (0.5 mM, 100 μL) were regarded as control groups ($n$ = 5). First, the accumulation of MNB-Pyra Nbs in vivo was evaluated by HPLC analysis[57,58]. The mice were sacrificed when the fluorescence signal reached peak value after the intravenous injection of MNB-Pyra Nbs (0.5 mM, 100 μL), and the organs and tumor tissues of same weight were prepared into tissue homogenates for HPLC analysis. As shown in Supplementary Fig. 28, the concentration of MNB-Pyra Nbs accumulated in tumor tissues was about 6.3 μM, which could cause efficient cell death under light compared to the IC$_{50}$ (0.2 μM under normoxia and 0.4 μM under hypoxia) of cells killing. Then, the mice for PDT treatment were exposed to light irradiation (630 nm, 50 mW/cm$^2$, 20 min) after the efficient accumulation of MNB-Pyra Nbs. As shown in Fig. 6b, in mice administered a single PDT treatment, tumor growth was not efficiently suppressed, with a tumor growth inhibition rate of only 18% (MNB-Pyra Nbs + L (1) group). However, the long-term retained fluorescence signal at the tumor site after light irradiation encouraged us to conduct continuous PDT on days 1, 2 and 3 (MNB-Pyra Nbs + L (3) group). Remarkably, the average tumor inhibition rate in this group was over 95%, and some tumors eventually disappeared after the mice were treated with three rounds of PDT. The tumor weights (Fig. 6c) and pictures of the tumors (Supplementary Fig. 29) from different groups at the end of the experiment also verified the highly effective inhibition provided by the MNB-Pyra Nbs after multiple PDT. These results clearly demonstrated that the MNB-Pyra Nbs successfully realized efficient tumor inhibition in a large-volume tumor model via sustained PDT after the administration of a single dose due to the precise tumor retention effect.

Terminal deoxynucleotidyl transferase-mediated dUTP nick-end labeling (TUNEL) staining was employed to label apoptotic cells. As anticipated, a significantly higher percentage of TUNEL-positive cells was found in the MNB-Pyra Nbs + L (3) group than in the other groups, indicating that the therapeutic efficacy in the MNB-Pyra Nbs + L (3) group was the highest due to the effective induction of apoptosis (Fig. 6e). Hematoxylin and eosin (H&E) staining was used to assess the morphology of the cells in tumor tissues. As depicted in Fig. 6f, a substantial number of tight stromal cells were observed in the control groups, and the intact cell nuclei indicated that the cells were in good condition. However, almost no cells with a complete morphology were observed in the MNB-Pyra Nbs + L (3) group, and the cell nuclei were shrunken or broken. Immunohistochemical (IHC) analysis also demonstrated the suppression of tumor proliferation with downregulated Ki-67 expression in the MNB-Pyra Nbs+L (3) group (Fig. 6g), whereas extensive cell damage was not observed in the other treatment groups. The physiological safety of MNB-Pyra Nbs in vivo was evaluated by monitoring the body weights of the mice and analyzing the health of the organs with H&E staining. Notably, none of the mice displayed any abnormal changes in body weight during the treatment period (Fig. 6d). Meanwhile, no noticeable cell necrosis or inflammatory lesions were observed in any of the major organs in all groups (Supplementary Fig. 30). These findings underscore the biocompatibility and applicability of MNB-Pyra Nbs in vivo.

Moreover, the therapeutic effect of MNB-Pyra Nbs in vivo was further evaluated on Hela tumor bearing model with moderate EGFR expression. The Hela tumor-bearing mouse model was constructed by subcutaneous injection of Hela cells into BALB/c nude female mice (5–6 weeks). First, the efficient targeting of MNB-Pyra to tumor tissue was confirmed in this model, and similar precise long-term retention of released photosensitizers after illumination was observed (Fig. 7a). The tumor inhibition efficiency of the conjugate was investigated in the large tumor model (initial volume $\approx$ 350 mm$^3$). Mice in different groups treated with only PBS, light, or MNB-Pyra Nbs (0.5 mM, 100 μL) were regarded as control groups ($n$ = 4). As shown in Fig. 7b, the tumor growth in MNB-Pyra Nbs + L (1) group (630 nm, 50 mW/cm$^2$, 20 min) was not efficiently suppressed with a tumor growth inhibition rate of only 23%. However, the average tumor inhibition rate in MNB-Pyra Nbs + L (3) group was about 90% after the mice were treated with three rounds of PDT. The tumor weights (Fig. 7c) and pictures of the tumors (Supplementary Fig. 31) from different groups at the end of the experiment also verified the highly-efficient tumor inhibition of MNB-Pyra Nbs after multiple PDT. H&E staining of tumor tissue displayed the broken cell morphology in MNB-Pyra Nbs + L (3) group only (Fig. 7e), and IHC analysis also declared downregulated Ki-67 expression in this group (Fig. 7f) indicating the suppression of tumor proliferation. Besides, the steady weight gain of mice weight (Fig. 7d) and healthy state of major organs (Supplementary Fig. 32) manifested the high physiological safety of MNB-Pyra Nbs. Consequently, MNB-Pyra Nbs exhibited effective inhibition to EGFR-overexpressed tumor in vivo.

## Discussion

PDT has been employed in the clinic for treatment of a range of cancers e.g. skin, lung, bladder, head and neck, and very recently primary breast cancer and non-oncological disorders (e.g. antimicrobial PDT, age-related macular degeneration)[4,59]. Even though the activation of the photosensitizers occurs only in the area where light is applied, the fact that conventional photosensitizers are hydrophobic, and non-selective molecules, makes PDT often associated with damage to surrounding normal tissue and unwanted phototoxicity side effects. The conjugation of more hydrophylic photosensitizers to conventional mAbs is currently being tested in the clinic and reduces these unwanted effects, by specifically targeting the photosensitizers to cancer cells[9,60]. However, the large size of mAbs limits their ability to penetrate tumors, restricting their effectiveness, and the generation of new or modified mAbs is costly and laborious[18]. These problems can be mitigated by the exploiting of Nbs, which is the smallest antigen-binding fragments with high stability and affinity derived from the variable domain of naturally occurring heavy-chain-only antibodies in camelids (VHH) and cartilaginous fish (VNAR). Sabrina Oliveira and her colleagues first conjugated Nbs with photosensitizers for nanobody-targeted PDT in 2014[23]. Subsequently, they utilized the nanobody-photosensitizer conjugate for the treatment of head and neck cancer and trastuzumab-resistant HER2-positive breast cancer[24,25]. And some other efforts were done to improve the therapeutic effect in nanobody-targeted PDT[26,28]. In these works, some progresses have been made but the photosensitizer used for conjugation is IRDye700DX only, while this hydrophilic photosensitizer still faces the limitation of tumor hypoxia environment during PDT. Meanwhile, the preparation of the conjugate requires multiple steps, and quantitative

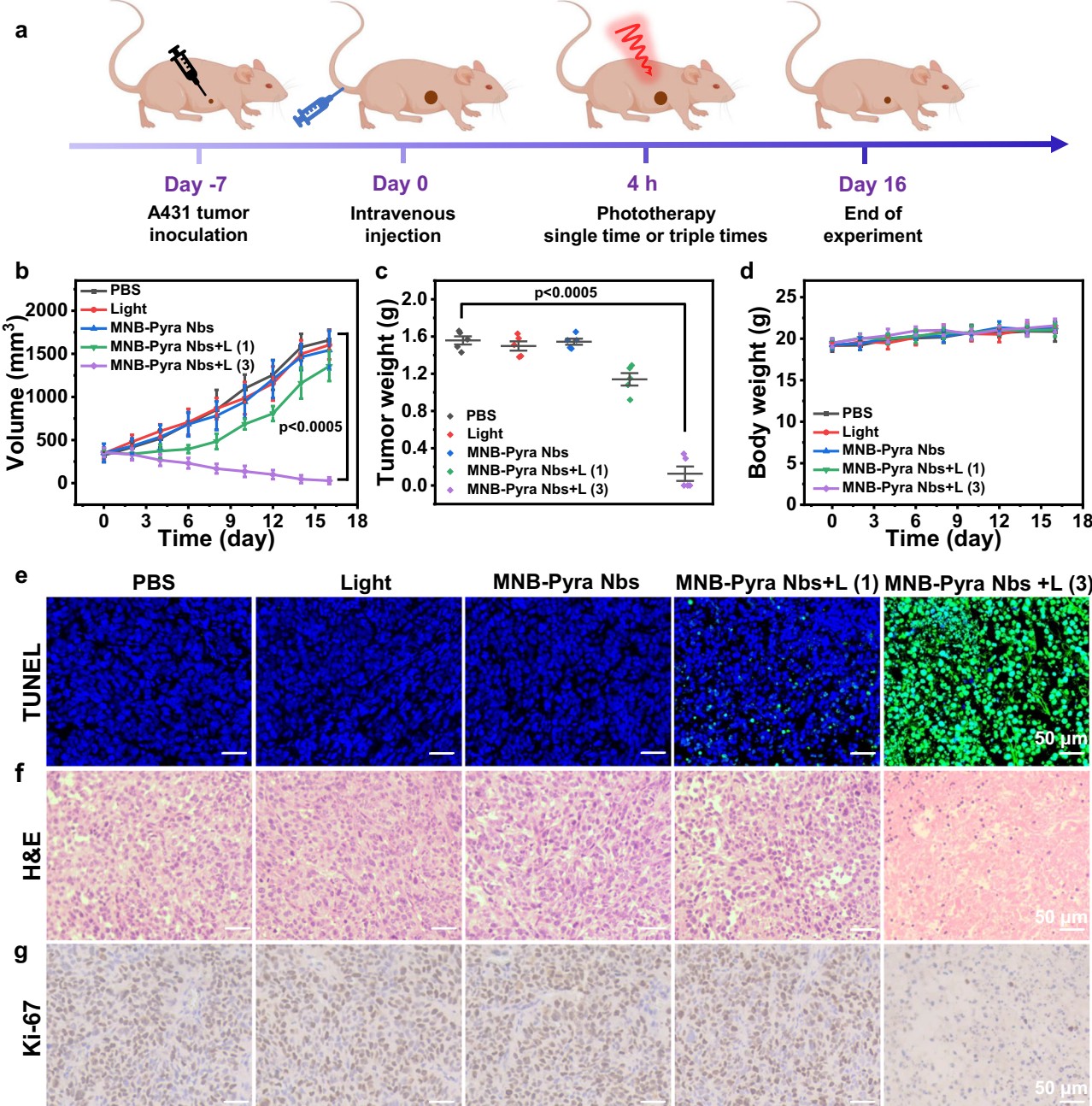

**Fig. 6 | The PDT effect of MNB-Pyra Nbs in A431 tumor bearing model.**
**a** Construction of large volume tumor model and therapeutic schedule for photodynamic therapy. Schematic created with Figdraw (ID: IIUTAd4f1d) and PowerPoint. **b** Changes of tumor volume in mice after different treatments (*n* = 5 mice per group). **c** Weight of tumors after 16 days in different groups (*n* = 5 mice per group). **d** Body weight changes in mice over 16 days after different treatments (*n* = 5 mice per group). **e** TUNEL assay analysis of tumor tissues after different treatments. **f** H&E analysis of tumors from different groups after treatments. **g** IHC analysis of Ki-67 from different groups. Data are shown as mean ± SD (**b**–**d**). Statistical significance in (**b**, **c**) was calculated via one tailed Student's *t* test. *P* < 0.05 is considered to be statistically significant. Scale bar = 50 μm. Source data are provided as a Source Data file.

modification at the specific site in Nbs cannot be easily achieved. Herein, the FGE modified 7D12-fGly with an aldehyde tag could realize specific and quantitative conjugation between the Nbs and type I photosensitizers via a Knoevenagel−Michael tandem reaction. The preparation of the conjugate was simple that Nbs and photosensitizers (1:10) only need to be added into a miniature reaction bottle to stir and the reaction yield is exceeding 95% in 24 h. Besides, the significantly increased fluorescence signal after illumination enables the conjugate to self-report the release of monomer photosensitizers and indicate the ROS production, which means such design functionalized this covalent conjugation between Nbs and photosensitizers.

Moreover, it is critical that photosensitizers have good targeting ability and improved tumor retention for accurate and effective PDT in vivo. However, Nbs are rapidly removed from mammals in few hours via the circulation because of their small size, which may cause insufficient accumulation in tumor tissue, especially in large volume tumors. Notably, photosensitizers demonstrating prolonged retention properties are well suited for the treatment of large-volume tumors, offering the advantages of reducing the pain caused by multiple dosing and minimizing the potential risks of side effects associated with repeated dosing[12]. Nevertheless, the drawback of prolonged retention is the challenge of clearing the photosensitizer from the body, which

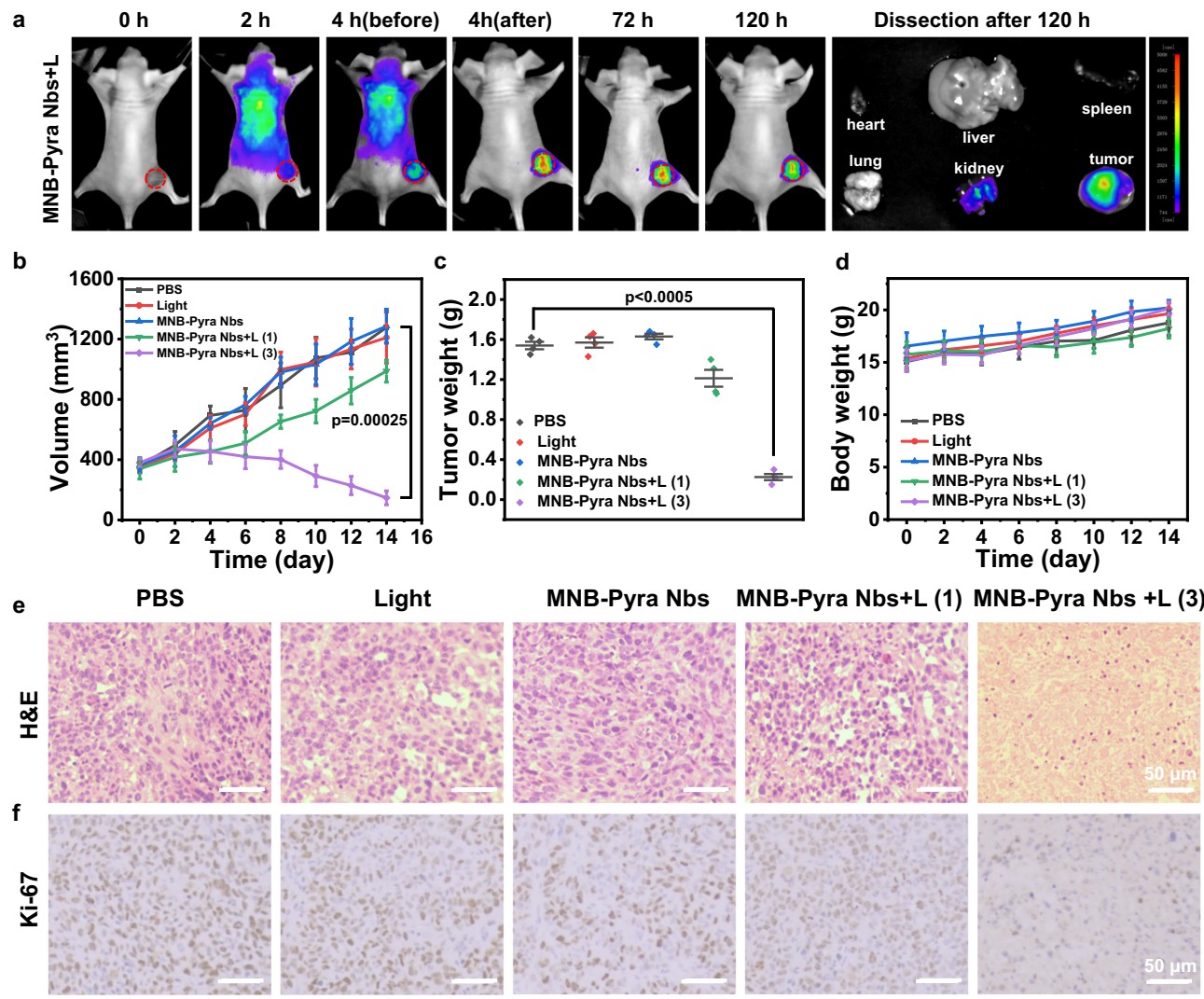

**Fig. 7 | PDT effect of MNB-Pyra Nbs in Hela tumor bearing model. a** Targeting and precise long-time retention imaging of MNB-Pyra Nbs in vivo. **b** Changes of tumor volume in mice after different treatments ($n = 4$ mice per group). **c** Weight of tumors after 14 days in different groups. ($n = 4$ mice per group) **d** Body weight changes in mice over 14 days after different treatments ($n = 4$ mice per group). **e** H&E analysis of tumors from different groups after treatments. **f** IHC analysis of Ki-67 from different groups. Data are shown as mean ± SD (**b**–**d**). Statistical significance in (**b**, **c**) was calculated via one tailed Student's *t* test. $P < 0.05$ is considered to be statistically significant. Scale bar = 50 μm. Source data are provided as a Source Data file.

may lead to phototoxicity in organs during treatment. A more strategic approach to mitigate this limitation is to precisely release long-term resident photosensitizers exclusively at the tumor site, facilitating sustainable PDT. Therefore, combining rapidly cleared Nbs with long-term retention photosensitizers through a cleavable chain offers the potential to enhance both the targeting capability of the molecules and the clearance of the long-term retention photosensitizers in normal tissues.

In this work, the photodynamic conjugates, MNB-Pyra Nbs were constructed by conjugating an EGFR-specific nanobody with the type I photosensitizer MNB-Pyra for sustainable PDT with efficient targeting ability. MNB-Pyra Nbs demonstrated potent toxicity to EGFR-positive cells under both normoxia and hypoxia conditions after PDT. Moreover, the specific modification of 7D12-fGly Nbs with MNB-Pyra in a 1:2 ratio realized significant fluorescence quenching, while the significant increase in fluorescence intensity after illumination could indicate the occurrence of PDT, making these modified Nbs self-reporting conjugates. Additionally, the covalent bond improved the clearance of the benzophenothiazine photosensitizer in vivo, leading to the clearance of MNB-Pyra Nbs in 24 h without illumination. MNB-Pyra Nbs displayed precise long-term tumor retention due to the release of the monomer

photosensitizer at the tumor site after light irradiation. This accurate accumulation and long-term retention of the photosensitizer was especially suitable for the ablation of large-volume tumors through sustained PDT after injection of a single dose. Finally, MNB-Pyra Nbs exhibited efficient tumor targeting and high efficiency PDT with tumor inhibition rates over 90% in vivo. This versatile and highly efficient photodynamic nanobody conjugate could provide a new perspective for the development of nanobody-targeted PDT. If nanobody-targeted PDT is to be further developed in the clinic, this conjugate will be directed towards investigating its precise long-term retention effect in preclinical models, to further determine the feasibility of the sustainable PDT.

## Methods

### Ethics statement

Our research complies with all relevant ethical regulations. The BALB/c nude female mice (5-6 weeks) used for animal studies were purchased from Beijing Vital River Laboratory Animal Technology Co., Ltd. All animal experiments were approved by the animal research ethics committee of Dalian University of Technology (approval number: DUT20210713). The animal study complied with relevant ethical

regulations for animal testing and research. The maximum tumor size/burden used in the present experiments was <1800 mm$^3$, which did not exceed the committee's maximal limit (2000 mm$^3$). Mice were housed in a specific pathogen-free (SPF) environment at the Laboratory Animal Center of Dalian University of Technology at a temperature of 22–26 °C, relative humidity of 50–60%, and alternating light and dark every 12 h.

## Materials

All chemical reagents used in this study were purchased from Energy Chemical Co., unless otherwise specified, and all solvents were of analytical grade. DHR 123 (dihydrorhodamine 123), DHE (Dihydroethidium), ABDA (9,10-Anthracenediyl-bis(methylene) dimalonic acid), MTT (3-(4,5-dimethyl-2-thiazolyl)-2,5-diphenyl-2H-tetrazolium bromide), Hoechst 33342, LysoTracker Green, Calcein-AM/propidium iodide (PI) Detection Kit, and Annexin V-FITC/PI Kit were purchased from Beyotime Biotechnology (Shanghai, China). Dulbecco's modified Eagle medium (DMEM) with L-glutamine was purchased from Solarbio. Penicillin–streptomycin (10000 U/mL), Trypsin–EDTA (0.05%), and fetal bovine serum (FBS) were purchased from Gibco. Silica gel (200–300 mesh) was used for flash-column chromatography.

## Mass spectrometry experiments

For organic molecules, electrospray ionization (EIS)-high resolution mass spectrometry (HRMS) analysis was performed with Synapt G2-Si HDMS (Waters, USA). The molecules were dissolved in methanol for test, and the concentration was 2 μM ($n = 1$). The collected data were analyzed by Masslynx software (Version 4.1, Waters, USA) and outputted as figures in supporting information. For 7D12 nanobody and the conjugate, HPLC–HRMS analysis was employed with LTQ Orbitrap XL (Thermo Scientific, USA). The nanobody or conjugate were dissolved in MES buffer for test ($n = 1$, 2 mg/mL). Liquid phase (LC) separation was performed using a Hypersil Gold™ C18 column. The LC separation conditions were acetonitrile (containing 0.1% formic acid)/water (containing 0.1% formic acid) by 5-70%. The LC flow rate was 0.5 mL/min and the operation time was 30 min. The protein coupling mass spectrometry data were deconvolution analyzed using the complete protein analysis module of BioPhamar Finder 2.0 software (Thermo finisher, USA).

## Computational details

The 2000 initial structures of MNB-Pyra dimer were generated using Genmer. The structures were then optimized using Molclus combined with GFN2-xTB. The redundant structures were removed with an energy threshold of 0.5 kcal/mol and structure threshold of 0.5 Å. Finally, structure with lowest energy (DE < = 3 kcal/mol) were kept for next optimization steps. Geometry optimization was carried out with B97-3c density functional method using Orca 5.0 software (Software update: The ORCA program system-Version 5.0). For MNB-Pyra dimer, the structures were first optimized using Molclus combined with orca 5.0 software. The redundant structures were then removed with an energy threshold of 0.5 kcal/mol and structure threshold of 0.5 Å. The single point energy of the structures remained were then calculated at the RI-wB97M-V/def2-TZVP level with SMD solvation model. The ratio of different structure was calculated according to Boltzmann distribution at 298.15 K.

## Cell lines

Human epidermal cancer cells (A431 cells; TCHu188), human cervical carcinoma cells (Hela cells; TCHu187), human non-small cell lung cancer cells (A549 cells; TCHu150) and mouse embryonic cells (NIH-3T3 cells; SCSP-5012) were purchased from the National Collection of Authenticated Cell Cultures (Shanghai, China). All cells were cultured in Dulbecco's modified Eagle's medium (DMEM;

C11995500BT, Gibco) supplemented with 10% (vol/vol) fetal bovine serum (FBS; 16000-044, Gibco) and 1% (vol/vol) penicillin/streptomycin solution (ST488-1/ST488-2, Beyotime). All cell lines were cultured at 37 °C in a humidified atmosphere containing 5% $CO_2$ and were regularly tested for the absence of mycoplasma and bacterial contamination.

## Cell-based enzyme linked immunosorbent assay

A431 and NIH-3T3 cellls (100 μL, $1 \times 10^3$ cells) were seeded into 96-well cell culture plates containing DMEM medium supplemented with 1% penicillin-streptomycin and 10% FBS and cultured overnight. Afterwards, the cells were washed three times with PBS solution. Then, 4% paraformaldehyde (125 μl) was added into each well and the cells were incubated at 37 °C for 30 min to fix the cells. The cells were then washed with PBS solution, and 5% (w/v) skim milk powder (200 μl) was added and the cells were incubated at 37 °C for 1 h. Next, 200 μL of 7D12-fGly and MNB-Pyra Nbs (30 μg/ml) were added and the cells were incubated at 37 °C for another 1 h. Subsequently, the cells were incubated with 200 μl of anti-6×His primary antibody solution (Sangon Biotech, China, diluted 1:2000 in PBS) for 1 h at 37 °C, and followed by incubation with a 200 μl solution of secondary HRP-conjugated Goat Anti-Mouse IgG antibody (Sangon Biotech, China, diluted 1:2000 in PBS) at 37 °C for 1 h. Finally, the cells were incubated with TMB working solution (200 μl) for 20 min, and followed by adding 50 μl of stop solution. $OD_{450}$ nm value of the resulting solution in the well was measured using a microplate reader.

## Subcellular localization

A431 cells ($1 \times 10^4$) were seeded into the cell dish and cultured overnight at 37 °C under a humidified atmosphere containing 5% $CO_2$. The cells were first incubated with MNB-Pyra Nbs (100 nM) for 2 h, and then Lyso-Tracker Green (1 μM, Beyotime Biotechnology) was added. The cells were incubated for another 30 min, followed by washing with PBS buffer three times. The cells were then imaged using CLSM under 488 nm and 640 nm excitation.

## Western blot

Cells ($5 \times 10^6$) were digested and washed with PBS three times, and the cells were resuspended in lysis buffer (50 mM Tris, 150 mM NaCl, 0.2% Triton X-100, 10 μg/ml PMSF, pH 8.0). After 20 min of lysis on ice, the protein concentration in the supernatant was determined using the BCA assay. 20 μg of total proteins were loaded on 10% SDS-PAGE. The proteins were then transferred to PVDF membranes, followed by blocking with 5% skim milk powder for 1 h. The membranes were then incubated with primary antibodies (1: 20,000 in 2% skim milk) (mouse EGFR monoclonal antibody, Proteintech) overnight at 4 °C, followed by washing with TBST buffer thrice. The membranes were further incubated with horseradish peroxidase-conjugated secondary antibody (1:20,000, goat anti-mouse IgG, proteintech) for 1 h at room temperature, and then were washed with TBST buffer thrice. The protein bands were visualized using ECL reagent under ChemiDoc XR+system (Bio-rad).

## Cell imaging

Cells ($5 \times 10^4$) were seeded into cell dishes containing DMEM supplemented with 1% penicillin-streptomycin and 10% FBS and cultured overnight. Various molecules were directly added to the cell culturing medium and followed by imaging using a confocal laser scanning microscope (Olympus FV3000) without washing ($\lambda_{ex} = 640$ nm, $\lambda_{em} = 690$–720 nm).

## Flow cytometry

Cells ($1 \times 10^5$) were incubated 100 nM of MNB-Pyra Nbs for 2 h, followed by digestion and washing with PBS. Cells were then resuspended

in PBS (200 µl) and analyzed using Attune™ NxT flow cytometer (ThermoFisher Scientific).

## Cell viability assays in vitro

Cell viability was measured by reducing of 3-(4,5)-dimethylthiahiazo(-2-yl)-3,5-diphenytetrazoliumromide (MTT) to formazan crystals under mitochondrial dehydrogenases activity in living cells. All types of cells were seeded in 96-well microplates and each well had a density of $1 \times 10^5$ cells per well in a 100 µL DMEM medium containing 10% FBS. After 24 h of cell growth, 100 µL different concentrations of molecule drugs from 0 to 2 µm in DMEM medium were added to the wells. The dark groups were continuously cultured for 12 h whereas the irradiation groups were irradiated with 630 nm light (30 mW/cm², 20 min) after 2 h of incubation with drugs and then cultured for another 12 h. Next, the MTT-containing medium was carefully poured out, and the generated formazan crystals were dissolved with 200 µL DMSO for each well. The microplate reader was taken to detect the absorbance of solutions at 490 nm and cell viability was calculated using the following equation:

$$\text{Cell viability}\,(\%) = [(\text{OD}_{\text{Ps}} - \text{OD}_{\text{blank control}})/(\text{OD}_{\text{control}} - \text{OD}_{\text{blank control}})] \times 100\% \tag{1}$$

Where the $\text{OD}_{\text{ps}}$ represents the absorbance of therapeutic group, $\text{OD}_{\text{blank control}}$ represents the absorbance of blank control group (without cells), and $\text{OD}_{\text{control}}$ represents the absorbance of control group.

## Tumor suppression experiments in vivo

Mice inoculated with A431 cells in armpit were divided into five groups randomly ($n = 4$ or $n = 5$) when their tumor volume reached about 350 mm³. The control groups were treated with saline, light or MNB-Pyra Nbs only. The fourth group was injected with MNB-Pyra Nbs (0.5 mM, 100 µL) treated with light (630 nm, 50 mW/cm², 20 min) only once, and the last group was injected with MNB-Pyra Nbs (0.5 mM, 100 µL) treated with light (630 nm, 50 mW/cm², 20 min) three time at the 1st, 2nd, and 3rd day respectively. The injection of saline or drugs was in an intravenous way and exposed under light irradiation after 4 h. Subsequently, the weight of the mice and the volume of primary and distant tumors were monitored every day for 15 days. The volume of the tumor was calculated using the following formula:

$$V = L \times D \times D/2 \tag{2}$$

where V is the tumor volume, L is the longest diameter of the tumor, and D is the diameter perpendicular to L.

## Statistics and reproducibility

All experiments were conducted at least twice independently, yielding similar results. Replicates were reproducible. Biologically independent samples/animals per group and the experimental data were analyzed by one-tailed Student's $t$ test. $P < 0.05$ was statistically significant. Experiments that related to micrographs were repeated at least three times and similar results were obtained. Blinding was used in all biological experiments and no data were excluded from the analysis.

## Reporting summary

Further information on research design is available in the Nature Portfolio Reporting Summary linked to this article.

## Data availability

The authors declare that the data supporting the findings of this study are available within the article and its Supplementary Information files. Source data are provided with this paper. All remaining data can be found in the Article, Supplementary and Source Data files. Source data are provided with this paper.

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

## Acknowledgements

This work was financially supported by National Natural Science Foun-dation of China (21925802, J.F., 22338005, J.F.), Liaoning Binhai Laboratory (LBLB-2023-03, J.F.) and the Fundamental Research Funds for the Central Universities (DUT22LAB601, J.F.).

## Author contributions

Y.C. designed the research. T.X., Q.P. performed the experiments. J.D., W.S., J.F., and X.P. analyzed the data, and wrote the paper.

## Competing interests

The authors declare no competing interests.
