## [Peer review file · Nature Communications]

Self-Reporting Photodynamic Nanobody Conjugate for Precise and Sustainable Large-Volume Tumor TreatmentREVIEWER COMMENTS

Reviewer #1 (Nanobody conjugates; EGFR inhibition):

The manuscript entitled: Self-Reporting Photodynamic Nanobody Conjugate for Precise and Sustainable Large-Volume Tumor Treatment, presents an interesting approach of quenched photosensitizer conjugated to nanobodies for targeted PDT.

Although several elements are quite interesting and certainly of relevance to the scientific community, there are a few aspects that limit the novelty and relevance for a publication to be granted in this highly prestigious journal. The authors are encouraged to adapt their manuscript so it can be considered for publication by possibly a more chemical or PDT oriented journal.

Authors are recommended to have more accurate explanations:

- Metabolism is often indicated while clearance is a more appropriate term.
- The definition of nanobody in the abstract should mention camelid origin and or heavy chain antibodies and that they are naturally occurring.
- The concept of active targeting may be well accepted by some readers, while not by others. Authors could reconsider explaining the targeting is through a specific interaction with a membrane receptor.

Authors are suggested to properly acknowledge the previous and specially the first work published on Nb targeted photodynamic therapy. Only 1 reference is indicated from that work, number 15, and actually not used correctly, as the sentence refers to the phase III clinical trial of cetuximab-PS.

“In this regard, accurately delivering long-term resident photosensitizers to the tumor site via Nbs may be an ideal strategy to reduce drug enrichment in normal tissues, enhance drug accumulation in tumor tissues, and enable precise and sustainable PDT, which offers the advantages of reducing the pain accompanied by multiple dosing and minimizing the potential risks of side effects associated with repeated dosing, especially for the treatment of large-volume tumors”. -> it would seem appropriate to here follow with evidences from published studies

“The nanobody 7D12 specifically binds to EGFR ectodomain epitopes inaccessible to mAbs, subsequently promoting EGFR aggregation and inducing receptor-mediated lysosomal endocytosis.^{22, 23}” -> this is not accurate: this NB alone binds to EGFR domain 3 and competes with cetuximab, so the 1st part is incorrect. Alone, it is taken up by fluid phase uptake, only when combined with other Nbs as biparatopic format or on nanoparticles it leads to EGFR clustering and is more efficiently internalized by receptor mediated endocytosis (correct terminology is preferred). See refs to support DOI: 10.1002/ijc.26145 , <http://dx.doi.org/10.1016/j.str.2013.05.008>; doi:10.1242/jcs.028753

The linker used is ROS sensitive, so part of the PS becomes free after illumination. So where are the ROS produced for the actual cell killing? What is the fate of the PS when released by light application? It is known that the Nb 7D12 leaves the tumor after some time (max achieved shortly after injection and by 2h p.i. already reduced, so how much PS is in fact delivered locally and of use after the first release by the ROS sensitive linker? Possibly another NB format (e.g. biparatopic) could remain longer in the mouse or at the tumor site. (<http://dx.doi.org/10.1016/j.jconrel.2016.03.014>)

Fig 2: shouldn't a difference of almost 2kDa result at least in a small shift of the band upwards? It now seems the small shift is to a lower Mw. Any explanation for this?

Approximately 15 nm (Figure 2d), which indicated that the particle size of the antibody did not change significantly after 7D12-fGly and MNB-Pyra were covalently bound.: antibody is here mentioned, should it be nanobody? Nanobodies have been indicated as 2.5 by 4 nm size. Could the authors comment on this or indicate more evidences of such size reported?

“Importantly, the binding affinity of the 7D12-fGly ligand for EGFR remained unaffected after conjugation, as confirmed by the cell-based ELISA results (Figure S5)”: no binding affinity is studied in this assay. This assay only shows that there is binding to positive cells and less binding to negative cells. A proper determination of (apparent) affinity with dilution series or, alternatively with SPR, would be needed.

NIH 3T3 is a murine fibroblast cell line, the others are human. HeLa have moderate expression, certainly not as high as A431. These should be corrected.

The authors could elaborate on how they envisage the re-illumination in the clinical setting.

Controls of light only should be included for each cell line.

Confirming their efficient PDT effect under hypoxia conditions.: what is the explanation for the mechanism?

Leading to the release of the monomer photosensitizer with increased fluorescence intensity to monitor the occurrence of PDT: this sentence seems odd as the ROS is consumed to release the PS. How much ROS is consumed for this and how much is in fact doing PDT? If not possible to measure accurately, an estimation would be important to add.

Not clear how the settings for illumination are chosen and why the same for in vitro and in vivo. 30mW/cm² for 20 min.

Unfortunately the model used in vivo is the A431 which is the easiest model to treat in that respect, due to the enormous amount of EGFR on the cell surface. Authors are encouraged to assess the potency of their treatment in models with more realistic or representative expression of EGFR.

The present discussion seems to be a small summary of what is presented, thus not discussing the current observations in light of previous published (very related) studies. Also, authors are encouraged to comment on the possible clinical translation.

Reviewer #2 (nanomedicine, cancer therapy):

Antibody-photosensitizer (PDT agent) conjugates are one of ADCs and have been reported in a lot of preclinical and clinical studies. In the sense, a nanobody (Nb)-photosensitizer conjugate reported in this article is conceptually not a new one. Despite that, the MNB-Pyra Nbs presented in this study contain several new and novel aspects, such as ROS-triggered release of a photosensitizer MNB-Pyra by introducing a thioketal linker (which is a well

know ROS cleavable linker, though) and subsequent long-term residence of the released PDT agent in the tumor which enables multiple PDT therapy effective for large tumor. The authors well demonstrated the proof of concept and the key claims proposed in this work with reasonable experimental setup and solid data. However, there are several parts that should be considered and revised to improve the overall quality of this paper.

Major points:

1. Although self-reporting (fluorescence 'ON') conjugate concept is interesting, it is unclear what kind of benefit can be obtained in the real clinical settings. The self-fluorescence quenched MNB-Pyra Nbs exhibit very little fluorescence, which means it is hard to see whether the drug conjugate is successfully localized in the tumor with sufficient amount for PDT. Rather, always 'ON' conjugate, mono MNB-Pyra Nbs (with the same ROS-cleavable linker) would be a better choice in that it can give when and how much of the drug conjugate is successfully localized in the target tumor. In the sense, mono MNB-Pyra Nbs should be tested and compared with MNB-Pyra Nbs.
2. Certain tumors produce high level of ROS in their microenvironment, for example, A549-derived tumor is one of them. It suggests that MNB-Pyra Nbs could release MNB-Pyra in such ROS-high tumors even without irradiation. This possibility should be tested by intratumoral injection of MNB-Pyra Nbs in such ROS-high and -low tumors.
3. In Figure 4b, the tumor accumulation of MNB-Pyra Nbs reached a maximum and gradually decreased with time after that peak time. It is likely that Nbs reached in the tumor would remain for long time after receptor-mediated endocytosis into cancer cells. If that is true, the accumulation of the Nb conjugates would increase with time. But the finding was opposite to the expectation. This finding should be explained and discussed.

Minor points:

1. Did the authors measure the hydrodynamic size of MNB-Pyra Nbs by DLS? Any change in aqueous solubility of MNB-Pyra Nbs compared to Nbs?
2. Typo in line 336 and 340: Not Fig. 5d, Fig. 5d and 5e, but Fig. 4d and Fig. 4d and 4e.
3. Define MNB-Pyra at its first appearance. And other abbreviations as well.

Reviewer #3 (PDT, photosensitizer):

Prof. Fan and his coworkers are presenting a nanobody-functionalized compound for PDT application. The nanobody acts as a delivery vector while a benzophenothiazine derivative plays the role of the photosensitizer. Together, the nanobody-conjugate reaches the target, and is activated in the presence of light to produce ROS, which kill cancer cells. In vitro and in vivo tests have demonstrated the potential of this hybrid system for PDT application.

Overall, in my opinion, the results can be published in Nature Communications after some minor revisions.

In the abstract, the Authors claimed personalized precision PDT (last sentence), however, what is personalized here ? For me, it seems to be quite general, and it should work for all patients. How can it be personalized ?

The characterization of the compounds, the photosensitizers, the dimers, etc... are performed in acetonitrile-methanol, far from biological conditions. Please provide some characterization in water, and if possible under physiological conditions (pH, buffer, 37°C, etc...).

According to Figure 2, and the text, the Authors suggest that the same structure of the dimer (with pi-pi stacking and quenching) is observed, at the periphery of the Nb. However, considering the size and potential at the surface of 7D12-tGly, and especially in water, are they sure about that. Another mechanism of quenching can probably take place.

Finally, regarding the stability of the conjugate, is it sensitive to enzymes, and other biomolecules ?

The quality of the photos in Figure 5 needs to be improved.

Reviewer #1 (Nanobody conjugates; EGFR inhibition):

The manuscript entitled: *Self-Reporting Photodynamic Nanobody Conjugate for Precise and Sustainable Large-Volume Tumor Treatment*, presents an interesting approach of quenched photosensitizer conjugated to nanobodies for targeted PDT.

Although several elements are quite interesting and certainly of relevance to the scientific community, there are a few aspects that limit the novelty and relevance for a publication to be granted in this highly prestigious journal. The authors are encouraged to adapt their manuscript so it can be considered for publication by possibly a more chemical or PDT oriented journal.

Response: Thanks for your comments and detail suggestions. To clearly express our novelty and make this study more detailed and convincing, we have made extensive modifications to our manuscript and implemented a bunch of additional experiments in the past several months according to your advice. The prolonged retention of photosensitizers can enable multiple PDT via single-dose injection with enhanced therapeutic outcomes [1] but it's also suggested that they are difficult to clear from the body, which brings about potential phototoxicity to organs during treatment and the risk of side effects caused by the overaccumulation of drugs. The high stability and affinity nanobodies bring a new opportunity to improve the specificity to tumors for photodynamic therapy (PDT). The precise delivery and release of long-time resident photosensitizers at the tumor site via Nbs after the cleavable chain connection was a novel and effective approach to reduce drug enrichment in normal tissues, enhance drug retention in tumor tissues, and enable precise and sustainable PDT, especially for the treatment of large-volume tumors. Meanwhile, the formylglycine-generating enzyme modified 7D12-fGly with an aldehyde tag realize specific and quantitative conjugation between the Nbs and type I photosensitizers in a 1:2 ratio via a Knoevenagel–Michael tandem reaction, in which the reaction yield is exceeding 95% in 24 h. Moreover, MNB-Pyra Nbs exhibited significant fluorescence quenching due to the π - π stacking interactions of the planar molecule MNB-Pyra after site-specific modification of the Nbs with MNB-Pyra in a 1:2 ratio, but the significantly increased fluorescence signal after illumination enables the conjugate to self-report the release of monomer photosensitizers and indicate the ROS production. We believe this versatile and highly

efficient photodynamic nanobody conjugate provides a paradigm for the design of precise long-time retention photosensitizers and is expected to promote the development of nanobody-targeted PDT. Besides, we have answered these important questions and supplied the related experiments and new validation data in the manuscript, marking with blue color.

References:

- [1]. Xi, D. et al. Strong pi-pi Stacking Stabilized Nanophotosensitizers: Improving Tumor Retention for Enhanced Therapy for Large Tumors in Mice. *Adv Mater* **34**, e2106797 (2022).

1. Authors are recommended to have more accurate explanations:

- (1) -Metabolism is often indicated while clearance is a more appropriate term.
- (2) -The definition of nanobody in the abstract should mention camelid origin and or heavy chain antibodies and that they are naturally occurring.
- (3) -The concept of active targeting may be well accepted by some readers, while not by others. Authors could reconsider explaining the targeting is through a specific interaction with a membrane receptor.

Response: Thanks for your advice.

- (1) We have replaced all the word “metabolism” with “clearance” in the manuscript.

The change in the manuscript:

In terms of the sustainability and phototoxic side effects of treatment, the cycling and **clearance** of photosensitizers in PDT must be considered. Excessively fast **clearance** of the photosensitizer results in insufficient drug accumulation, hindering subsequent sustained treatment, while slow **clearance** leads to a long drug clearance period, potentially causing severe phototoxic side effects.³⁰⁻³²

- (2) We have revised the definition of antibody in the abstract.

The change in the manuscript:

Nanobodies (Nbs), the smallest antigen-binding fragments with high stability and affinity **derived from the variable domain of naturally occurring heavy-chain-only antibodies in camelids**, bring a new opportunity to improve the specificity to tumors for photodynamic therapy (PDT).

(3) We have revised the description of this targeting mechanism in the manuscript.

The change in the manuscript:

To achieve high tumor specificity, antibody–drug conjugates (ADCs) has garnered lots of attention as a targeting approach with high biosafety due to the specific interaction between antibodies and cell membrane receptors.

2. Authors are suggested to properly acknowledge the previous and specially the first work published on Nb targeted photodynamic therapy. Only 1 reference is indicated from that work, number 15, and actually not used correctly, as the sentence refers to the phase III clinical trial of cetuximab-PS.

Response: Thanks for your advice. In this paragraph, the first thing we want to describe is that antibody–drug conjugates (ADCs) has garnered lots of attention and monoclonal antibodies currently play a major role in ADCs development. However, monoclonal antibody still faces some problems and the advantages of nanobody can solve these problems. Then, the content of nanobody is introduced in the second half of this paragraph. We are sorry for not citing the correct reference here and we have corrected the related references of the phase III clinical trial of cetuximab-photosensitizer. We also revised our manuscript for a better understanding. Meanwhile, previous and the first published work on Nb targeted photodynamic therapy were added in this paragraph.

The change in the manuscript:

To achieve high tumor specificity, antibody–drug conjugates (ADCs) has garnered lots of attention as a targeting approach with high biosafety due to the specific interaction between antibodies and cell membrane receptors. At present, the development of ADCs is mainly based on monoclonal antibodies (mAbs) conjugates, including the antibody–photosensitizer conjugates in PDT.^{13,14} For example, the conjugation of cetuximab mAb and IRdye700DX photosensitizer, has been extensively studied for the treatment of recurrent head and neck cancer.^{9,15} However, the challenges associated with specific conjugation and the high production costs of mAbs have limited the development of ADCs.¹⁶⁻¹⁸ In contrast, nanobodies (Nbs), the smallest antigen-binding fragments derived from heavy chain-only antibodies that are exclusively found in camelids (VHH) and cartilaginous fish (VNAR), have gained much interest.^{19,20} Their single-domain

nature allows their direct expression in bacterial systems, making them easier to prepare and modify as needed. Compared to conventional mAbs with a size of approximately 150 kDa, the small size of Nbs (approximately 15 kDa) enables them to reach less accessible antigens while maintaining high affinity and stability.^{21,22} Sabrina Oliveira and her colleagues first conjugated Nbs with photosensitizers for nanobody-targeted PDT in 2014,²³ and some other nanobody-targeted PDT related works were published.²⁴⁻²⁷ Nevertheless, Nbs are rapidly removed from mammals via the circulation with an elimination half-life of several hours due to their small size.^{17,28,29}

The change in the references:

9. Mitsunaga, M. et al. Cancer cell-selective in vivo near infrared photoimmunotherapy targeting specific membrane molecules. *Nat. Med.* **17**, 1685-1691 (2011).
15. Okada, R. et al. The Effect of Antibody Fragments on CD25 Targeted Regulatory T Cell Near-Infrared Photoimmunotherapy. *Bioconjugate Chemistry* **30**, 2624-2633 (2019).
23. Heukers, R., van Bergen en Henegouwen, P.M.P. & Oliveira, S. Nanobody–photosensitizer conjugates for targeted photodynamic therapy. *Nanomedicine: Nanotechnology, Biology and Medicine* **10**, 1441-1451 (2014).
24. van Driel, P. et al. EGFR targeted nanobody-photosensitizer conjugates for photodynamic therapy in a pre-clinical model of head and neck cancer. *J. Control. Release.* **229**, 93-105 (2016).
25. Deken, M.M. et al. Nanobody-targeted photodynamic therapy induces significant tumor regression of trastuzumab-resistant HER2-positive breast cancer, after a single treatment session. *J. Control. Release.* **323**, 269-281 (2020).
26. Mashayekhi, V., Xenaki, K.T., van Bergen en Henegouwen, P.M.P. & Oliveira, S. Dual Targeting of Endothelial and Cancer Cells Potentiates In Vitro Nanobody-Targeted Photodynamic Therapy. *Cancers* **12**, 2732 (2020).
27. Beltrán Hernández, I. et al. The Potential of Nanobody-Targeted Photodynamic Therapy to Trigger Immune Responses. *Cancers* **12**, 978 (2020).

3. “In this regard, accurately delivering long-term resident photosensitizers to the tumor site via Nbs may be an ideal strategy to reduce drug enrichment in normal tissues, enhance drug accumulation in tumor tissues, and enable precise and sustainable PDT, which offers the advantages of reducing the pain accompanied by multiple dosing and minimizing the potential risks of side effects associated with repeated dosing, especially for the treatment of large-volume tumors”. -> it would seem appropriate to here follow with evidences from published studies.

Response: Thanks for your advice. This sentence is an ideal design proposed by us to

solve the problems caused by the excessively fast clearance or slow clearance of photosensitizers. However, this sentence may be too long and complicated to understand easily. Thus, we modified the sentence to make it more concise.

The change in the manuscript:

In this regard, accurately delivering long-term resident photosensitizers to the tumor site via Nbs may be an ideal strategy to reduce drug enrichment in normal tissues, enhance drug retention in tumor tissues, and enable precise and sustainable PDT, especially for the treatment of large-volume tumors.

4. “The nanobody 7D12 specifically binds to EGFR ectodomain epitopes inaccessible to mAbs, subsequently promoting EGFR aggregation and inducing receptor-mediated lysosomal endocytosis.^{22, 23}” -> this is not accurate: this NB alone binds to EGFR domain 3 and competes with cetuximab, so the 1st part is incorrect. Alone, it is taken up by fluid phase uptake, only when combined with other Nbs as biparatopic format or on nanoparticles it leads to EGFR clustering and is more efficiently internalized by receptor mediated endocytosis (correct terminology is preferred). See refs to support DOI:10.1002/ijc.26145, <http://dx.doi.org/10.1016/j.str.2013.05.008>; doi:10.1242/jcs.028753

Response: Thanks for your advice. We have revised this description in the manuscript according to these references.

The change in the manuscript:

The nanobody 7D12 specifically binds to the ligand-binding domain III of the EGFR and is subsequently endocytosed into the cells by fluid phase uptake.^{18,38,39}

The change in the references:

18. Schmitz, Karl R., Bagchi, A., Roovers, Rob C., van Bergen en Henegouwen, Paul M.P. & Ferguson, Kathryn M. Structural Evaluation of EGFR Inhibition Mechanisms for Nanobodies/VHH Domains. *Structure* **21**, 1214-1224 (2013).
38. Roovers, R.C. et al. A biparatopic anti-EGFR nanobody efficiently inhibits solid tumour growth. *International Journal of Cancer* **129**, 2013-2024 (2011).
39. Hofman, E.G. et al. EGF induces coalescence of different lipid rafts. *Journal of Cell Science*

5. (1) The linker used is ROS sensitive, so part of the PS becomes free after illumination. So where are the ROS produced for the actual cell killing? (2) What is the fate of the PS when released by light application? (3) It is known that the Nb 7D12 leaves the tumor after some time (max achieved shortly after injection and by 2h p.i. already reduced, so how much PS is in fact delivered locally and of use after the first release by the ROS sensitive linker? (4) Possibly another NB format (e.g. biparatopic) could remain longer in the mouse or at the tumor site. (<http://dx.doi.org/10.1016/j.jconrel.2016.03.014>)

Response: Thanks for your advice.

(1) The distribution of released free photosensitizers was investigated in Supplementary Figure 15b. The result showed that the released photosensitizers were still localized in the lysosomes of the cells after illumination. Moreover, the ROS generating situation of released photosensitizers was also evaluated (Supplementary Figure 15c), which confirmed the ROS for cell killing were produced inside the cells.

The change in the manuscript:

Furthermore, the released photosensitizers after illumination were also located in the lysosomes and could product ROS efficiently inside cells with re-illumination (Supplementary Figure 15b and c).

Supplementary Figure 15. (b) Co-localization of released free photosensitizers in A431 cells. (c) ROS production of the released photosensitizers with fluorescence probe DCFH-DA (E_x : 488 nm, E_m : 500-550 nm) in A431 cells. Scale bar = 20 μ m.

(2) To analyze the fate of released photosensitizers *in vivo*, we further investigated the long-term imaging results in A431-bearing tumor model (Supplementary Figure 24). The result suggested that the released photosensitizers at the tumor site would be cleared slowly through the kidney way in about 12 days. This drug clearance at this tumor site is different from the systemic circulatory clearance of long-term resident drugs because its drug concentrations are much lower and there is little risk of side effects through the kidney way only. Moreover, during therapy, part of the long-term resident photosensitizers at the tumor site will be removed as the scab falls off as shown in the following picture. Therefore, this strategy of precise long-term retention via Nbs provides a more convenient and safer way to reduce the drug enrichment in normal tissues but enhance drug retention in tumor tissue for the achievement of sustainable PDT after a single dose of administration.

The change in the manuscript:

The released monomer photosensitizers at the tumor site would be cleared in about 12 days (Supplementary Figure 24). This drug clearance at this tumor site is different from the systemic circulatory clearance of long-term resident drugs because its drug concentrations are much lower and there is little risk of side effects through the kidney way only. Therefore, this strategy of precise long-term retention via Nbs provides a more convenient and safer way to reduce the drug enrichment in normal tissues but enhance drug retention in tumor tissue for the achievement of sustainable PDT after a single dose of administration.

Supplementary Figure 24. Monitoring the clearance of the photosensitizers after released by light application.

(3) The accumulation of MNB-Pyra Nbs in tumor tissue was evaluated by high-performance liquid chromatography (HPLC) analysis. The mice were sacrificed when the fluorescence signal reached peak value after the intravenous injection of MNB-Pyra Nbs (0.5 mM, 100 μ L), and the organs and tumor tissues of same weight were prepared into tissue homogenates for HPLC analysis. [2,3] As shown in Supplementary Figure 26, the concentration of MNB-Pyra Nbs accumulated in tumor tissue was about 6.3 μ M, which is a very high concentration of drug compared to the IC_{50} (0.2 μ M under normoxia and 0.4 μ M under hypoxia) of cells killing though the drug enrichment rate is not high. This concentration was enough to cause efficient cell death under light while also had high biosafety in the absence of light (Figure 3b). Moreover, the release efficiency of MNB-Pyra Nbs was investigated by the absorption spectrum in Supplementary Figure 12, in which the release efficiency was about 83% after light irradiation. Therefore, the concentration of the monomer photosensitizers in an ideal way could be about 10 μ M at the tumor site after illumination.

References:

- [2]. Sahin, S. et al. Preparation, characterization and in vivo distribution of terbutaline sulfate loaded albumin microspheres. *Journal of Controlled Release* **82**, 345-358 (2002).
- [3]. Shen, G. et al. Tissue distribution of 2-methoxyestradiol nanosuspension in rats and its antitumor activity in C57BL/6 mice bearing lewis lung carcinoma. *Drug Delivery* **19**, 327-333 (2012).

The change in the manuscript:

First, the accumulation of MNB-Pyra Nbs *in vivo* was evaluated by HPLC analysis.^{57,58} The mice were sacrificed when the fluorescence signal reached peak value after the intravenous injection of MNB-Pyra Nbs (0.5 mM, 100 μ L), and the organs and tumor

tissues of same weight were prepared into tissue homogenates for HPLC analysis. As shown in Supplementary Figure 26, the concentration of MNB-Pyra Nbs accumulated in tumor tissues was about 6.3 μM , which could cause efficient cell death under light compared to the IC_{50} (0.2 μM under normoxia and 0.4 μM under hypoxia) of cells killing. Then, the mice for PDT treatment were exposed to light irradiation (630 nm, 50 mW/cm^2 , 20 min) after the efficient accumulation of MNB-Pyra Nbs.

Supplementary Figure 26. (a) Standard curve of MNB-Pyra Nbs (1,5,10,30,50 μM) obtained by HPLC. (b) Drug concentrations in organs and tumor 4 hours after intravenous injection of MNB-Pyra Nbs (0.5 mM, 100 μL) in mice. Data are shown as mean \pm SD ($n = 3$).

The release rate of MNB-Pyra Nbs after illumination was investigated by the absorption spectrum in Supplementary Figure 12, which exhibited a high release efficiency about 83%.

Supplementary Figure 12. Absorbance of MNB-Pyra Nbs (5 μM) after light irradiation in DMEM compared with that of MNB-Pyra (10 μM) to calculate the release efficiency of monomer photosensitizers.

(4) The Nb format such as biparatopic may be an efficient way to prolong the circulation of Nbs in vivo, but this kind of prolonged retention is different from the precise long-term retention in this work. Herein, the rapid clearance of Nbs in vivo improved the clearance of long-time resident photosensitizer in organs and avoid severe phototoxicity side effects during therapy. Meanwhile, the long-time resident photosensitizers were only released at the tumor site after illumination to achieve sustainable PDT with high biosafety. We aim to prolong the retention of photosensitizers at the tumor site only but not to prolong the circulation of the conjugate in vivo. Of course, the conjugation of biparatopic Nbs with photosensitizers may improve the accumulation of drugs due to the higher internalization capacity of Nbs, which needs to be explored further.

6. Fig 2: shouldn't a difference of almost 2kDa result at least in a small shift of the band upwards? It now seems the small shift is to a lower Mw. Any explanation for this?

Response: Actually, the charge and hydrophobicity of nanobody would change after conjugation due to the small size of Nbs (15kDa) and the positive charge property of photosensitizers, so it is normal that the band shift situation does not meet prediction. The similar situations have been observed in our previous work. [4] Therefore, the conjugate should be further characterized by high-performance liquid chromatography-high resolution mass spectrometry (HPLC-HRMS) to confirm its formation (Figure 2c).

References:

[4]. Peng, Q., Xiong, T., Ji, F., Ren, J. & Jia, L. Reduction-Activatable Fluorogenic Nanobody for Targeted and Low-Background Bioimaging. *Analytical Chemistry* **95**, 2804-2811 (2023).

7. Approximately 15 nm (Figure 2d), which indicated that the particle size of the antibody did not change significantly after 7D12-fGly and MNB-Pyra were covalently bound.: antibody is here mentioned, should it be nanobody? Nanobodies have been indicated as 2.5 by 4 nm size. Could the authors comment on this or indicate more evidences of such size reported?

Response: Thanks for your advice. We did want to discuss the size of the nanobody here, so we changed the word as nanobody. To further verify the size of the conjugate, the hydrodynamic size was investigated by dynamic light scattering (DLS). It was

found that the hydrodynamic diameter of 7D12-fGly Nbs and MNB-Pyra Nbs was 5.3 nm and 6.1 nm, respectively (Figure 2d). This result is more reliable and in line with previous reports. As for the TEM images, we originally wanted to refer to the characterization method of nanomedicines and observe the morphology of the particles after dropping the conjugate substance on the copper web (diameter 300 μm) with natural drying. However, the large particle sizes in TEM may be attributed to protein aggregation and precipitation after sample preparation, because the TEM sample is placed at room temperature for more than 15 hours to wait for natural air drying. Therefore, the particle sizes reported by TEM was not accurate and we replaced the TEM images with the DLS result.

The change in the manuscript:

The dynamic light scattering (DLS) analysis showed that the hydrodynamic diameter of 7D12-fGly Nbs and MNB-Pyra Nbs was 5.3 nm and 6.1 nm respectively (Figure 2d), suggesting the particle size of the nanobody did not change significantly after the conjugation.

Figure 2. (d) Hydrodynamic size of 7D12-fGly and MNB-Pyra Nbs by dynamic light scattering.

8. “Importantly, the binding affinity of the 7D12-fGly ligand for EGFR remained unaffected after conjugation, as confirmed by the cell-based ELISA results (Figure S5)”: no binding affinity is studied in this assay. This assay only shows that there is binding to positive cells and less binding to negative cells. A proper determination of (apparent) affinity with dilution series or, alternatively with SPR, would be needed.

Response: Thanks for your advice. We investigated the binding affinity of 7D12-fGly Nbs and MNB-Pyra Nbs with biolayer interferometry (BLI), which is also a valuable

method for measuring protein-protein and protein-small molecule interactions. [5-7] As shown in Supplementary Figure 8, 7D12-fGly Nbs had high bind affinity to EGFR with a dissociation constant (K_D) value of 1.79 ± 0.27 nM and MNB-Pyra Nbs had a dissociation constant (K_D) value of 24.33 ± 0.65 nM to EGFR. This decrease of binding affinity after conjugation is acceptable because the coupled molecules (2kD) had a considerable proportion in the scale of MNB-Pyra Nbs (19kD), thereby the introduction of molecules would affect the binding affinity of Nbs to EGFR. However, MNB-Pyra Nbs exhibited EGFR binding capacity comparable to 7D12-fGly Nbs in the cell-based ELISA analysis (Supplementary Figure 9). Therefore, the binding of the 7D12-fGly ligand for EGFR remained unaffected after conjugation though the dissociation constant (K_D) value slight decreased.

References:

- [5]. Nirschl, M., Reuter, F. & Vörös, J. Review of Transducer Principles for Label-Free Biomolecular Interaction Analysis. *Biosensors* **1**, 70-92 (2011).
- [6]. Hou, S., Sun, P., Zhang, M. & Han, B. Capturing the Interaction Kinetics of an Ion Channel Protein with Small Molecules by the Bio-layer Interferometry Assay. *Journal of Visualized Experiments* (2018).
- [7]. Tanaka, K. et al. Development of oligonucleotide-based antagonists of Ebola virus protein 24 inhibiting its interaction with karyopherin alpha 1. *Organic & Biomolecular Chemistry* **16**, 4456-4463 (2018).

The change in the manuscript:

The binding affinity of 7D12-fGly Nbs and MNB-Pyra Nbs to EGFR were evaluated by biolayer interferometry (BLI), which analyzes biomolecular interactions via an optical biosensing technology.⁴⁸⁻⁵⁰ As shown in Supplementary Figure 8, 7D12-fGly Nbs and MNB-Pyra Nbs showed comparable EGFR binding affinity, which have nanomolar range of dissociation constant (K_D) value of 1.79 ± 0.27 nM and 24.33 ± 0.65 nM, respectively. Though the introduction of photosensitizers slightly decreased the binding affinity of Nbs, the conjugate still exhibited significant EGFR binding capacity in the cell-based ELISA analysis (Supplementary Figure 9). Therefore, it is convinced that the Nbs can reserve its high binding affinity after the site-specific conjugation, achieving the delivery of photosensitizers toward EGFR overexpressed

cells.

Supplementary Figure 8. Binding affinity evaluation of (a) 7D12-fGly Nbs and (b) MNB-Pyra Nbs with biolayer interferometry.

9. NIH 3T3 is a murine fibroblast cell line, the others are human. Hela have moderate expression, certainly not as high as A431. These should be corrected.

Response: Thanks for your advice. We have corrected these contents in the manuscript.

The change in the manuscript:

Therefore, four different cell lines with different EGFR expression levels were employed for cell experiments, including human epidermoid carcinoma (A431), human cervical carcinoma (HeLa), human lung cancer (A549) and mouse embryonic cells (NIH-3T3). Western blot assays confirmed that A431 and HeLa cells are EGFR-positive cells, in which A431 exhibited high EGFR expression and HeLa had moderate expression, while A549 and NIH-3T3 cells had almost negligible EGFR expression (Figure 3a).

10. The authors could elaborate on how they envisage the re-illumination in the clinical setting.

Response: Thanks for your advice. In fact, the re-illumination process has no difference from the common PDT treatment in the clinical, which only needs to repeat the illumination process of conventional PDT. At present, PDT in clinical mainly achieves illumination in two ways. First, for the skin or superficial lesions, direct illumination with LED or low-dose laser can be repeated multiple times after the efficient accumulation of photosensitizers. Second, for the non-superficial tumors, light sources

currently can be transported through optical fibers in clinical practice to realize illumination. In this case, repeated optical fiber introduction or intervention is required for multiple treatments to achieve tumor inhibition. [8,9]

References:

- [8]. van Straten, D., Mashayekhi, V., de Bruijn, H.S., Oliveira, S. & Robinson, D.J. Oncologic Photodynamic Therapy: Basic Principles, Current Clinical Status and Future Directions. *Cancers (Basel)* **9** (2017).
- [9]. Zou, J. et al. Evaluation of lensed fibers used in photodynamic therapy (PDT). *Photodiagnosis and Photodynamic Therapy* **31**, 101924 (2020).

11. Controls of light only should be included for each cell line.

Response: Thanks for your advice. We have supplemented the light only group in related experiments.

The change in the manuscript:

Supplementary Figure 16. ROS generating detection of MNB-Pyra Nbs in A431 cells under normoxia (21% O₂) and hypoxia (2% O₂) conditions with fluorescence probe DCFH-DA (E_x: 488 nm, E_m: 500-550 nm) and DHE (E_x: 560 nm, E_m: 600-630nm). Scale bar = 20 μm.

Supplementary Figure 18. Live/death cells imaging of different cells with or without PDT treatment (630 nm, 30 mW/cm², 20 min) using Calcein-AM/PI kit. Calcein-AM: Ex: 488 nm, Em: 490–520 nm; PI: Ex: 488 nm, Em: 600–700 nm. Scale bar = 200 μm.

Supplementary Figure 20. Apoptosis imaging of A431 cells using Annexin V-FITC (E_x: 488 nm, E_m: 490–530 nm) and PI (E_x: 488 nm, E_m: 600–700 nm) kit. Scale bar = 20 μm.

Supplementary Figure 21. Apoptosis detection using flow cytometry in A431 cells.

12. Confirming their efficient PDT effect under hypoxia conditions.: what is the explanation for the mechanism?

Response: In general, based on the different photochemical mechanisms of action, PDT can be classified into type I PDT and type II PDT. In a type II mechanism, the excited triplet state photosensitizer can directly transfer its energy to nearby molecular O_2 , forming reactive singlet oxygen (1O_2) species. As this process needs the continued participation of O_2 , type II PDT is usually known as strongly O_2 dependence. In a type I mechanism, which is also known as an electron transfer process, an electron (or proton) is transferred from or to the excited triplet state photosensitizer to or from the biological substrate to generate a radical cation and corresponding anion. These radicals are unstable and capable of instantly reacting with surrounding O_2 to create reactive oxygen radicals like superoxide radicals ($O_2^{\cdot-}$). Via disproportionation reaction and Haber–Weiss/Fenton reaction, part of the generated $O_2^{\cdot-}$ leads to the cycle of O_2 and the formation of hydroxyl radicals ($\cdot OH$) with higher cytotoxicity to damage tumor cells. Therefore, type I PDT is low O_2 -dependent, and its efficiency is less affected by the concentration of O_2 . [10,11]

In our work, this benzophenothiazine photosensitizer MNB-Pyra has been proved as a type I photosensitizer, which efficiently produces superoxide radicals ($O_2^{\cdot-}$) by electron transfer after photoexcitation (Figure 1g,1h and 3e), so MNB-Pyra Nbs had efficient PDT effect under hypoxia conditions.

References:

- [10]. Li, M. et al. Near-Infrared Light-Initiated Molecular Superoxide Radical Generator: Rejuvenating Photodynamic Therapy against Hypoxic Tumors. *J Am Chem Soc* **140**, 14851-

14859 (2018).

[11]. Li, M., Xu, Y., Peng, X. & Kim, J.S. From Low to No O₂-Dependent Hypoxia Photodynamic Therapy (hPDT): A New Perspective. *Accounts of Chemical Research* **55**, 3253-3264 (2022).

13. Leading to the release of the monomer photosensitizer with increased fluorescence intensity to monitor the occurrence of PDT: this sentence seems odd as the ROS is consumed to release the PS. How much ROS is consumed for this and how much is in fact doing PDT? If not possible to measure accurately, an estimation would be important to add.

Response: Thanks for your advice. We apologize for the ambiguity of this sentence caused by the lack of clear expression. In fact, we want to emphasize that the increased fluorescence signal after illumination can reflect the generation of ROS and release of monomer photosensitizers, which can also indicate the occurrence of PDT but it's difficult to estimate the amount of ROS consumed for the cleavage and the amount of ROS in fact doing PDT.

The change in the manuscript:

Thus, it was suggested that the MNB-Pyra Nbs were cleaved in the cells after illumination, leading to the release of the monomer photosensitizer accompanied with fluorescence increasement (Figure 3n). Meanwhile, the enhanced fluorescence signal also feedbacks the generation of ROS and indicates the occurrence of PDT.

14. Not clear how the settings for illumination are chosen and why the same for in vitro and in vivo. 30mW/cm² for 20 min.

Response: In our work, we used LED light sources for illumination. The benzophenothiazine photosensitizer is a kind of highly efficient photosensitizer. According to previous related work, the light power of this kind of photosensitizer for cell killing in vitro was usually used as 20 mW/cm², 5-30 min and for tumor inhibition in vivo usually ranged from 50-100 mW/cm², 10-30 min. [12-15] Herein, considering that the ROS generated under light irradiation will be used to cleave the conjugate and kill cells, we increased the optical density to 30 mW/cm² to ensure the efficient generation of ROS. The light power used for sustainable PDT in vivo was (50 mW/cm²,

20 min), and we forgot to mention it in the main text. Besides, the light power of 30 mW/cm² was used in the fluorescence imaging in vivo because we only focus on its light-activated properties here to observe the fluorescence increasement and precise long-term retention of released photosensitizers at the tumor site after illumination.

References:

- [12]. Lu, Y. et al. Cancer immunogenic cell death via photo-pyroptosis with light-sensitive Indoleamine 2,3-dioxygenase inhibitor conjugate. *Biomaterials* **278**, 121167 (2021).
- [13]. Huang, D. et al. A Tumor-Specific Platform of Peroxynitrite Triggering Ferroptosis of Cancer Cells. *Advanced Functional Materials* **32** (2022).
- [14] Li, M. et al. Superoxide Radical Photogenerator with Amplification Effect: Surmounting the Achilles' Heels of Photodynamic Oncotherapy. *J Am Chem Soc* **141**, 2695-2702 (2019).
- [15]. Xiong, T. et al. Lipid Droplet Targeting Type I Photosensitizer for Ferroptosis via Lipid Peroxidation Accumulation. *Advanced Materials* **36** (2023).

The change in the manuscript:

Then, the mice for PDT treatment were exposed to light irradiation (630 nm, 50 mW/cm², 20 min) after the efficient accumulation of MNB-Pyra Nbs.

15. Unfortunately, the model used in vivo is the A431 which is the easiest model to treat in that respect, due to the enormous amount of EGFR on the cell surface. Authors are encouraged to assess the potency of their treatment in models with more realistic or representative expression of EGFR.

Response: Thanks for your advice. HeLa cells have a moderate expression of EGFR, so we constructed a HeLa tumor-bearing mouse model in Nu/J female mice to evaluate the therapeutic effect of MNB-Pyra Nbs in vivo again. As shown in Figure 6b, the tumor growth in MNB-Pyra Nbs + L (1) group (630 nm, 50 mW/cm², 20 min) was not efficiently suppressed with a tumor growth inhibition rate of only 23%. However, the average tumor inhibition rate in MNB-Pyra Nbs + L (3) group was about 90% after the mice were treated with three rounds of PDT. The tumor weights (Figure 6c) and pictures of the tumors (Supplementary Figure 29) from different groups at the end of the experiment also verified the highly-efficient tumor inhibition of MNB-Pyra Nbs after multiple PDT. H&E staining of tumor tissue displayed the broken cell morphology in

MNB-Pyra Nbs + L (3) group only (Figure 6f), and IHC analysis also declared downregulated Ki-67 expression in this group (Figure 6f) indicating the suppression of tumor proliferation. Besides, the steady weight gain of mice weight (Figure 6d) and healthy state of major organs (Supplementary Figure 30) manifested the high physiological safety of MNB-Pyra Nbs. Consequently, MNB-Pyra Nbs exhibited effective inhibition to EGFR-overexpressed tumor in vivo.

The change in the manuscript:

Moreover, the therapeutic effect of MNB-Pyra Nbs in vivo was further evaluated on HeLa tumor bearing model with moderate EGFR expression. The HeLa tumor-bearing mouse model was constructed by subcutaneous injection of HeLa cells into Nu/J female mice. First, the efficient targeting of MNB-Pyra to tumor tissue was confirmed in this model, and similar precise long-term retention of released photosensitizers after illumination was observed (Figure 6a). The tumor inhibition efficiency of the conjugate was investigated in the large tumor model (initial volume $\approx 350 \text{ mm}^3$). Mice in different groups treated with only PBS, light, or MNB-Pyra Nbs (0.5 mM, 100 μL) were regarded as control groups (n=4). As shown in Figure 6b, the tumor growth in MNB-Pyra Nbs + L (1) group (630 nm, 50 mW/cm², 20 min) was not efficiently suppressed with a tumor growth inhibition rate of only 23%. However, the average tumor inhibition rate in MNB-Pyra Nbs + L (3) group was about 90% after the mice were treated with three rounds of PDT. The tumor weights (Figure 6c) and pictures of the tumors (Supplementary Figure 29) from different groups at the end of the experiment also verified the highly-efficient tumor inhibition of MNB-Pyra Nbs after multiple PDT. H&E staining of tumor tissue displayed the broken cell morphology in MNB-Pyra Nbs + L (3) group only (Figure 6f), and IHC analysis also declared downregulated Ki-67 expression in this group (Figure 6f) indicating the suppression of tumor proliferation. Besides, the steady weight gain of mice weight (Figure 6d) and healthy state of major organs (Supplementary Figure 30) manifested the high physiological safety of MNB-Pyra Nbs. Consequently, MNB-Pyra Nbs exhibited effective inhibition to EGFR-overexpressed tumor in vivo.

Figure 6. The PDT effect of MNB-Pyra Nbs in HeLa tumor bearing model. (a) Targeting and precise long-time retention imaging of MNB-Pyra Nbs in vivo. (b) Changes of tumor volume in mice after different treatments. (c) Weight of tumors after 14 days in different groups. (d) Body weight changes in mice over 14 days after different treatments. (e) H&E analysis of tumors from different groups after treatments. (f) IHC analysis of Ki-67 from different groups. Data are shown as mean \pm SD (n = 4). ***P < 0.001 determined by Students' t-test. Scale bar = 50 μ m.

Supplementary Figure 29. Tumor picture at the end of experiment in different groups in HeLa tumor bearing model. Scale bar = 2 cm.

Supplementary Figure 30. H&E analysis of the major organs (heart, liver, spleen, lung, and kidney) tissues collected from mice in the different groups at the end of treatment in Hela tumor bearing model. Scale bar = 200 μ m.

16. The present discussion seems to be a small summary of what is presented, thus not discussing the current observations in light of previous published (very related) studies. Also, authors are encouraged to comment on the possible clinical translation.

Response: Thanks for your advice. We have revised the discussion part in the text.

The change in the manuscript:

PDT has been employed in the clinic for treatment of a range of cancers (*e.g.* skin, lung, bladder, head and neck, and very recently primary breast cancer and non-oncological disorders (*e.g.* antimicrobial PDT, age-related macular degeneration)).^{4,59} Even though the activation of the photosensitizers occurs only in the area where light is applied, the fact that conventional photosensitizers are hydrophobic, and non-selective molecules, makes PDT often associated with damage to surrounding normal tissue and unwanted phototoxicity side effects. The conjugation of more hydrophilic photosensitizers to conventional mAbs is currently being tested in the clinic and reduces these unwanted effects, by specifically targeting the photosensitizers to cancer cells.^{9,60} However, the large size of mAbs limits their ability to penetrate tumors, restricting their effectiveness, and the generation of new or modified mAbs is costly and laborious.¹⁸ These problems can be mitigated by the exploiting of Nbs, which is the smallest

antigen-binding fragments with high stability and affinity derived from the variable domain of naturally occurring heavy-chain-only antibodies in camelids (VHH) and cartilaginous fish (VNAR). Sabrina Oliveira and her colleagues first conjugated Nbs with photosensitizers for nanobody-targeted PDT in 2014.²³ Subsequently, they utilized the nanobody-photosensitizer conjugate for the treatment of head and neck cancer and trastuzumab-resistant HER2-positive breast cancer.^{24,25} And some other efforts were done to improve the therapeutic effect in nanobody-targeted PDT.^{26,28} In these works, some progresses have been made but the photosensitizer used for conjugation is IRDye700DX only, which is a kind of conventional photosensitizers facing the limitation of tumor hypoxia environment during PDT. Meanwhile, the preparation of the conjugate requires multiple steps, and quantitative modification at the specific site in Nbs cannot be easily achieved. Herein, the FGE modified 7D12-fGly with an aldehyde tag could realize specific and quantitative conjugation between the Nbs and type I photosensitizers via a Knoevenagel–Michael tandem reaction. The preparation of the conjugate was simple, in which Nbs and photosensitizers (1:10) only need to be added into a miniature reaction bottle to stir and the reaction yield is exceeding 95% in 24 h. Besides, the significantly increased fluorescence signal after illumination enables the conjugate to self-report the release of monomer photosensitizers and indicate the ROS production, which means such design functionalized this covalent conjugation between Nbs and photosensitizers.

Moreover, it is critical that photosensitizers have good targeting ability and improved tumor retention for accurate and effective PDT in vivo. However, Nbs are rapidly removed from mammals in few hours via the circulation because of their small size, which may cause insufficient accumulation in tumor tissue, especially in large volume tumors. Notably, photosensitizers demonstrating prolonged retention properties are well suited for the treatment of large-volume tumors, offering the advantages of reducing the pain caused by multiple dosing and minimizing the potential risks of side effects associated with repeated dosing.⁶¹ Nevertheless, the drawback of prolonged retention is the challenge of clearing the photosensitizer from the body, which may lead to phototoxicity in organs during treatment. A more strategic approach to mitigate this limitation is to precisely release long-term resident photosensitizers exclusively at the

tumor site, facilitating sustainable PDT. Therefore, combining rapidly cleared Nbs with long-term retention photosensitizers through a cleavable chain offers the potential to enhance both the targeting capability of the molecules and the clearance of the long-term retention photosensitizers in normal tissues.

In this work, the photodynamic conjugates, MNB-Pyra Nbs were constructed by conjugating an EGFR-specific nanobody with the type I photosensitizer MNB-Pyra for sustainable PDT with remarkable targeting ability. MNB-Pyra Nbs demonstrated potent toxicity to EGFR-positive cells under both normoxia and hypoxia conditions after PDT. Moreover, the specific modification of 7D12-fGly Nbs with MNB-Pyra in a 1:2 ratio realized significant fluorescence quenching, while the significant increase in fluorescence intensity after illumination could indicate the occurrence of PDT, making these modified Nbs self-reporting conjugates. Additionally, the covalent bond greatly improved the clearance of the benzophenothiazine photosensitizer in vivo, leading to the clearance of MNB-Pyra Nbs in 24 h without illumination. MNB-Pyra Nbs displayed precise long-term tumor retention due to the release of the monomer photosensitizer at the tumor site after light irradiation. This accurate accumulation and long-term retention of the photosensitizer was especially suitable for the ablation of large-volume tumors through sustained PDT after injection of a single dose. Finally, MNB-Pyra Nbs exhibited excellent tumor targeting and high efficiency PDT with tumor inhibition rates over 90% in vivo. This versatile and highly efficient photodynamic nanobody conjugate could provide a new perspective for the development of nanobody-targeted PDT. If nanobody-targeted PDT is to be further developed in the clinic, this conjugate will be directed towards investigating its precise long-term retention effect in preclinical models, to further determine the feasibility of the sustainable PDT.

Reviewer #2 (nanomedicine, cancer therapy):

Antibody-photosensitizer (PDT agent) conjugates are one of ADCs and have been reported in a lot of preclinical and clinical studies. In the sense, a nanobody (Nb)-photosensitizer conjugate reported in this article is conceptually not a new one. Despite that, the MNB-Pyra Nbs presented in this study contain several new and novel aspects, such as ROS-triggered release of a photosensitizer MNB-Pyra by introducing a thioketal linker (which is a well know ROS cleavable linker, though) and subsequent long-term residence of the released PDT agent in the tumor which enables multiple PDT therapy effective for large tumor. The authors well demonstrated the proof of concept and the key claims proposed in this work with reasonable experimental setup and solid data. However, there are several parts that should be considered and revised to improve the overall quality of this paper.

Response: Thanks for your comments and suggestions about the experiments, we have supplied related experimental results and corrected the problems in the text. The changes in the manuscript were marked with blue color.

Major points:

1. Although self-reporting (fluorescence “ON”) conjugate concept is interesting, it is unclear what kind of benefit can be obtained in the real clinical settings. The self-fluorescence quenched MNB-Pyra Nbs exhibit very little fluorescence, which means it is hard to see whether the drug conjugate is successfully localized in the tumor with sufficient amount for PDT. Rather, always “ON” conjugate, mono MNB-Pyra Nbs (with the same ROS-cleavable linker) would be a better choice in that it can give when and how much of the drug conjugate is successfully localized in the target tumor. In the sense, mono MNB-Pyra Nbs should be tested and compared with MNB-Pyra Nbs.

Response: Thanks for your advice. Drug overdose and excessive radiation induce side effects and lesions of normal tissues. Even worse, they can increase the risk of developing cancer resistance and second cancers, and aggravate the situation. [1] The accurate evaluation of where and when the photosensitizer is working and on-demand dosage is of great significance to precision treatment, as well as minimizing toxicity and side effects. [2,3] However, additional fluorescent probes need to be introduced

into most current PDTs, which makes the monitoring process complicated and delayed. Therefore, the development of self-reporting photosensitizers is of great significance in avoiding excessive damage and reducing toxic side effects.

As for studying the properties of mono MNB-Pyra Nbs, on one hand, it is known that the aldehyde group in 7D12-fGly would bind to small molecules in a 1:2 ratio based on previous references. [4] So we want to take advantage of this feature to achieve intramolecular stacking and realize fluorescence “OFF-ON”. On the other hand, unfortunately, the mono MNB-Pyra Nbs is not available because this Knoevenagel–Michael tandem reaction is uncontrollable. Even if the reaction time is shortened and the mixture of single substitution and double substitution is obtained, it is difficult to separate these two conjugates due to the small scale of the reaction (200 μ L) and the similar molecular mass (compared to the 17kD of nanobody). But high purity double substituted conjugates can be obtained by extending the reaction time easily, and protein HPLC–HRMS analysis confirmed the conjugation efficiency is exceeding 95% in 24 h, which this product is ready for use after several times of washing and ultrafiltration.

We believe the “OFF-ON” photosensitizers are of great significance for the diagnosis and treatment of diseases to improve the accuracy and the signal-to-noise ratio. Meanwhile, to avoid the disadvantages caused by the weak fluorescence after fluorescence quenching, it is more advantageous to develop ratio fluorescence molecules or multimodal imaging. [2,5]

References:

- [1]. Zhang, T. et al. In Situ Monitoring Apoptosis Process by a Self-Reporting Photosensitizer. *Journal of the American Chemical Society* **141**, 5612-5616 (2019).
- [2]. Wang, C. et al. Self-Immolative Photosensitizers for Self-Reported Cancer Phototheranostics. *Journal of the American Chemical Society* **145**, 13099-13113 (2023).
- [3]. Miao, J. et al. Heavy Atom-Free, Mitochondria-Targeted, and Activatable Photosensitizers for Photodynamic Therapy with Real-Time In-Situ Therapeutic Monitoring. *Angewandte Chemie International Edition* **61** (2022).
- [4]. Kudirka, R.A. et al. Site-Specific Tandem Knoevenagel Condensation-Michael Addition To Generate Antibody-Drug Conjugates. *ACS Med Chem Lett* **7**, 994-998 (2016).

[5]. Guan, K. et al. A two-photon fluorescence self-reporting black phosphorus nanoprobe for the in situ monitoring of therapy response. *Chemical Communications* **56**, 14007-14010 (2020).

2. Certain tumors produce high level of ROS in their microenvironment, for example, A549-derived tumor is one of them. It suggests that MNB-Pyra Nbs could release MNB-Pyra in such ROS-high tumors even without irradiation. This possibility should be tested by intratumoral injection of MNB-Pyra Nbs in such ROS-high and -low tumors.

Response: Thanks for your advice. We constructed an A549 and an A431 tumor-bearing mouse model in Nu/J female mice. Then, MNB-Pyra Nbs (0.3 mM, 100 μ L) was intratumorally injected and the fluorescence signals at tumor sites were collected at 0 h and 2 h. As a result, no obvious fluorescence increasements were observed in both A549 and A431 tumor models (Supplementary Figure 25), which indicated the ROS level in tumor microenvironment is insufficient to cleave this connecting chain.

The change in the manuscript:

Notably, the level of ROS in cancer cells is much higher than normal cells due to their increased metabolism and mitochondrial dysfunction.^{55,56} In order to confirm whether the level of ROS in cancer cells will cleave the conjugate, MNB-Pyra Nbs (0.3 mM, 200 μ L) was intratumoral injected in an A549 and an A431 tumor-bearing mouse model and the fluorescence signals at tumor sites were collected after 2 h. And no obvious fluorescence increasements were observed in both A549 and A431 tumor models (Supplementary Figure 25), which indicated the ROS level in tumor microenvironment is insufficient to cleave this connecting chain.

Supplementary Figure 25. Fluorescence observation after intratumoral injection of MNB-Pyra Nbs in A431 and A549 tumor models.

3. In Figure 4b, the tumor accumulation of MNB-Pyra Nbs reached a maximum and gradually decreased with time after that peak time. It is likely that Nbs reached in the tumor would remain for long time after receptor-mediated endocytosis into cancer cells. If that is true, the accumulation of the Nb conjugates would increase with time. But the finding was opposite to the expectation. This finding should be explained and discussed.

Response: Thanks for your advice.

In fact, nanobodies (Nbs) are rapidly removed from the circulation of mammals with an elimination half-life of several hours. [6] It is predominantly caused by the small molecular size of Nbs (15 kDa) as compared to intact IgG (150 kDa), resulting in their passage of the glomerular filter in the kidney and disposal in the urine. [7] Furthermore, due to Nbs are the smallest antigen-binding fragments lacking of the Fc fragment (Fc domain), Nbs are unable to interact with Fc receptor neonatal (FcRn, a receptor molecule located on the surface of cell membranes, which can bind to IgG-like antibodies). This receptor protects intact antibodies from catabolism by binding them after pinocytosis of blood by vascular epithelium and recycling them into the circulation. The importance of this mechanism was shown by transgenic mice lacking this receptor, which had a 10-fold decreased in antibody half-life. [8,9] In our work, we aim to use Nbs to improve drug clearance in organs and realize precise release of long-term retention photosensitizers only at the tumor site after illumination due to its excellent tumor targeting ability.

References:

- [6]. Harmsen, M.M., Van Solt, C.B., Fijten, H.P. & Van Setten, M.C. Prolonged in vivo residence times of llama single-domain antibody fragments in pigs by binding to porcine immunoglobulins. *Vaccine* **23**, 4926-4934 (2005).
- [7]. Batra, S.K., Jain, M., Wittel, U.A., Chauhan, S.C. & Colcher, D. Pharmacokinetics and biodistribution of genetically engineered antibodies. *Current Opinion in Biotechnology* **13**, 603-608 (2002).
- [8]. Lin, J.H., Guo, Y. & Wang, W. Challenges of Antibody Drug Conjugates in Cancer Therapy: Current Understanding of Mechanisms and Future Strategies. *Current Pharmacology Reports* **4**, 10-26 (2018).
- [9]. Sapra, P. et al. Anti-CD74 Antibody-Doxorubicin Conjugate, IMMU-110, in a Human Multiple Myeloma Xenograft and in Monkeys. *Clinical Cancer Research* **11**, 5257-5264 (2005).

Minor points:

1. Did the authors measure the hydrodynamic size of MNB-Pyra Nbs by DLS? Any change in aqueous solubility of MNB-Pyra Nbs compared to Nbs?

Response: Thanks for your advice. We evaluated the hydrodynamic size of Nbs by DLS, and the result in Figure 2d manifested the average hydrodynamic diameter of 7D12-fGly Nbs and MNB-Pyra Nbs was 5.3 nm and 6.1 nm, respectively.

Due to the hydrophobicity of small molecules, the water solubility of Nbs will be influenced after the conjugation. However, a 7D12-fGly nanobody is linked to only two molecules, which have little effect on its water solubility. This result was confirmed by the picture in Supplementary Figure 6, which a high concentration of MNB-Pyra Nbs (0.5 mM, 100 μ L) in aqueous solution remained clear and transparent without solid precipitation after being placed at 4°C for 5 days.

The change in the manuscript:

The dynamic light scattering (DLS) analysis showed that the hydrodynamic diameter of 7D12-fGly Nbs and MNB-Pyra Nbs was 5.3 nm and 6.1 nm respectively (Figure 2d), suggesting the particle size of the nanobody did not change significantly after the conjugation.

Figure 2. (d) Hydrodynamic size of 7D12-fGly and MNB-Pyra Nbs by dynamic light scattering.

MNB-Pyra Nbs exhibited good water solubility, which the high concentration of MNB-Pyra Nbs (0.5 mM, 100 μ L) in aqueous solution remained clear and transparent without solid precipitation after being placed at 4°C for 5 days (Supplementary Figure 6).

Supplementary Figure 6. Stability of MNB-Pyra Nbs (0.5 mM, 100 μ L) in aqueous solution after being placed at 4°C for 5 days.

2. Typo in line 336 and 340: Not Fig. 5d, Fig. 5d and 5e, but Fig. 4d and Fig. 4d and 4e.

Response: Thanks for your advice. We have corrected the errors in the manuscript.

3. Define MNB-Pyra at its first appearance. And other abbreviations as well.

Response: Thanks for your advice. We explained this abbreviation at its first appearance and checked other abbreviations in the manuscript.

The change in the manuscript:

The benzothiazine structure of morpholine-modified Nile blue (MNB) has been classified as a type I photosensitizer, which efficiently produces superoxide radicals ($O_2^{\cdot-}$) by electron transfer after photoexcitation.^{40, 41} Then, the MNB-Pyra molecule was constructed to covalently modify the Nbs by connecting MNB with the pyrazolone (Pyra) structure through a ROS-cleavable linker.^{42, 43}

Reviewer #3 (PDT, photosensitizer):

Prof. Fan and his coworkers are presenting a nanobody-functionalized compound for PDT application. The nanobody acts as a delivery vector while a benzophenothiazine derivative plays the role of the photosensitizer. Together, the nanobody-conjugate reaches the target, and is activated in the presence of light to produce ROS, which kill cancer cells. In vitro and in vivo tests have demonstrated the potential of this hybrid system for PDT application.

Overall, in my opinion, the results can be published in Nature Communications after some minor revisions.

Response: Thanks for your comments and detail proofreading, we have supplied the revised figures and experiments in the manuscript, marking with blue color.

1. In the abstract, the Authors claimed personalized precision PDT (last sentence), however, what is personalized here? For me, it seems to be quite general, and it should work for all patients. How can it be personalized?

Response: Photosensitizers to precise target and change fluorescence upon light illumination could accurately self-report where and when the photosensitizers work, enabling us to visualize the therapeutic process and precisely regulate treatment outcomes, which is the unremitting pursuit of precision and personalized medicine. [1, 2] Ideally, the self-reporting process could monitor the treatment process and provide guidance for drug dosage during treatment to minimize unwanted toxicity and side effects. Therefore, the real-time monitoring of photosensitizers without additional fluorescent probes provides the basis for personalized medicine.

In fact, we prefer to emphasize the achievement of precise long-term retention in vivo by this cleavable nanobody conjugate, and the self-reporting ability is an additional attribute. Thus, to clarify the innovation of this work, we revised the summary sentence of the abstract.

References:

- [1]. Zhang, T. et al. In Situ Monitoring Apoptosis Process by a Self-Reporting Photosensitizer. *J. Am. Chem. Soc.* **141**, 5612-5616 (2019).
- [2]. Wang, C. et al. Self-Immolative Photosensitizers for Self-Reported Cancer Phototheranostics. *J. Am. Chem. Soc.* **145**, 13099-13113 (2023).

The change in the manuscript:

The self-reporting conjugate provides a paradigm for the design of precise long-time retention photosensitizers and is expected to promote the development of PDT.

2. The characterization of the compounds, the photosensitizers, the dimers, etc... are performed in acetonitrile-methanol, far from biological conditions. Please provide some characterization in water, and if possible, under physiological conditions (pH, buffer, 37°C, etc...).

Response: Thanks for your advice. We investigated the absorbance and fluorescence spectra of MNB-Pyra and MNB-Pyra dimer in PBS solution (Figure 1b and c). The MNB-Pyra dimer exhibited a blue-shifted band at 600 nm and significant fluorescence quenching compared with MNB-Pyra. Furthermore, the ROS generation ability of the compounds was also evaluated in aqueous solution. As shown in Supplementary Figures 3 and 4, both the MNB-Pyra molecule and MNB-Pyra dimer exhibited negligible $^1\text{O}_2$ generation and efficient $\text{O}_2^{\cdot-}$ generation in PBS solution, which is consistent with previous results.

The change in the manuscript:

Figure 1. (b) Absorbance and (c) fluorescence spectra of MNB-Pyra (5 μM) and MNB-Pyra dimer (5 μM) in PBS solution.

1,3-Diphenylisobenzofuran (DPBF) and 9,10-Anthracenediyl-bis(methylene) dimalonic acid (ABDA) were employed to detect the generation of $^1\text{O}_2$, where dihydroethidium (DHE) and dihydrorhodamine 123 (DHR 123) served as indicators of $\text{O}_2^{\cdot-}$ with fluorescence emission at 600 nm and 526 nm. As anticipated, both the MNB-Pyra and MNB-Pyra dimers exhibited negligible $^1\text{O}_2$ generation (Figures 1f and Supplementary Figure 3) and efficient $\text{O}_2^{\cdot-}$ generation (Figures 1g and Supplementary Figure 4), affirming their excellent efficiency as type I photosensitizers.

Supplementary Figure 3. (a) Detection of $^1\text{O}_2$ generation by MNB-Pyra and MNB-Pyra dimer using DPBF probe in MeOH solution. (b) Detection of $^1\text{O}_2$ generation by MNB-Pyra and MNB-Pyra dimer using ABDA probe in aqueous solution.

Supplementary Figure 4. (a) $\text{O}_2^{\cdot-}$ generation of MNB-Pyra and MNB-Pyra dimer detected by DHE probe with ctDNA in PBS solution. (b) $\text{O}_2^{\cdot-}$ generation of MNB-Pyra and MNB-Pyra dimer detected by DHR123 probe in PBS solution.

3. According to Figure 2, and the text, the Authors suggest that the same structure of the dimer (with pi-pi stacking and quenching) is observed, at the periphery of the Nb. However, considering the size and potential at the surface of 7D12-fGly, and especially in water, are they sure about that. Another mechanism of quenching can probably take

place.

Response: The conjugate was washed and ultrafiltered several times after preparation to remove the unbound small molecules, and mass spectrometry characterization showed that each protein linked two molecules. Since two molecules bind at the same site resulting in the formation of dimer, DCQ is the most likely fluorescence quenching mechanism. However, the positive electric nature of the molecule does have the possibility of aggregation due to the adsorption with proteins, so we further studied the concentration-dependent absorption spectra of the conjugate. As displayed in Supplementary Figure 10a, MNB-Pyra molecules exhibit intermolecular aggregation in aqueous solution, which is affected by molecular concentration and shows obvious blue shift with increasing concentration. However, the increased single peak of blue-shifted absorption bands was observed in the absorbance spectra of MNB-Pyra dimer (Supplementary Figure 10b) and MNB-Pyra Nbs (Supplementary Figure 10c) with increasing concentration, where the aggregation degree is independent of their concentration. This result emphasizes the fluorescence quenching in the conjugate is same to the MNB-Pyra dimer, which is the intramolecular aggregation effect through DCQ, rather than the concentration-dependent intermolecular aggregation. [3]

References:

- [3]. Aparin, I.O. et al. Fluorogenic Dimers as Bright Switchable Probes for Enhanced Super-Resolution Imaging of Cell Membranes. *J. Am. Chem. Soc.* **144**, 18043-18053 (2022).

The change in the manuscript:

What's more, the increased single peak of blue-shifted absorption bands was observed in the absorbance spectra of MNB-Pyra dimer and MNB-Pyra Nbs with increasing concentration (Supplementary Figure 10), where the aggregation degree is independent of their concentration. This result declared the aggregation mechanism of DCQ in MNB-Pyra Nbs.

Supplementary Figure 10. Concentration-dependent (3-24 μM) absorbance spectra of (a) MNB-Pyra and (b) MNB-Pyra dimer in water. (c) Concentration-dependent (2-10 μM) trace absorbance spectra of MNB-Pyra Nbs.

4. Finally, regarding the stability of the conjugate, is it sensitive to enzymes, and other biomolecules?

Response: The stability of the conjugate could be reflected by the fluorescence intensity after the co-incubation with enzymes/other biomolecules. As shown in Supplementary Figure 7, this conjugate displayed high stability with negligible fluorescence changes after the co-incubation with enzymes/other biomolecules.

The change in the manuscript:

Besides, the negligible fluorescence change of the conjugate after the co-incubation with enzymes/other biomolecules reflected its high stability under physiological conditions (Supplementary Figure 7).

Supplementary Figure 7. 1. Fluorescence of MNB-Pyra in PBS solution. Fluorescence intensity of MNB-Pyra Nbs in the presence of enzymes/other biomolecules (2: aminopeptidases, 50 ng/mL; 3: γ -glutamyltransferase, 50 mU/mL; 4: alkaline phosphatase, 50 ng/mL; 5: nitroreductase, 10 $\mu\text{g/mL}$; 6: esterase, 50 mU/mL; 7: glutathione; 8: cysteine; 9: tyrosine; 10: glycine; 11: glutamate; 12:

arginine; 13: tryptophan; 14: aspartic acid; 15: serine; 1 mM).

5. The quality of the photos in Figure 5 needs to be improved.

Response: Thanks for your advice. We have reprocessed these photos to enlarge the scale and improved its sharpness.

The change in the manuscript as below:

REVIEWER COMMENTS

Reviewer #1 (Remarks to the Author):

The authors have addressed most of the points raised by this reviewer. Still, a few small adjustments for finalization are encouraged:

Nanobodies (Nbs), the smallest antigen-binding fragments with high stability and affinity derived from the variable domain of naturally occurring heavy-chain-only antibodies in camelids, bring a new opportunity to improve the specificity to tumors for photodynamic therapy (PDT) -> this is not the novel aspect of this article. So here should be correctly phased e.g. already shown to be an opportunity to improve... PDT.

Moreover, during therapy, part of the long-term resident photosensitizers at the tumor site will be removed as the scab falls off as shown in the following picture. -> after illumination the PS is likely not functional anymore, as usually photobleaching indicates its exhaustion/full use. The authors could add clarification on this aspect.

The Nb format such as biparatopic may be an efficient way to prolong the circulation of Nbs in vivo -> that is not the case, there is no substantial prolongation of half-life with biparatopics, compared to monovalent. Later it is indicated the half-life of monovalent Nbs is of several hours but usually 60-90 min are referred in literature.

Sabrina Oliveira and her colleagues first conjugated Nbs with photosensitizers for nanobody-targeted PDT in 2014.²³ -> they first published on this approach in 2014.

In these works, some progresses have been made but the photosensitizer used for conjugation is IRDye700DX only, which is a kind of conventional photosensitizers... -> this is a hydrophilic PS much different than most commonly used PS, so this sentence is misleading.

Reviewer #2 (Remarks to the Author):

Although this revision seems to address some of my concerns well, there are still unclear responses to two of major comments raised previously. Thus, another revision is necessary.

1. My comment "Certain tumors produce high level of ROS in their microenvironment, for example, A549-derived tumor is one of them. It suggests that MNB-Pyra Nbs could release MNB-Pyra in such ROS-high tumors even without irradiation. This possibility should be tested by intratumoral injection of MNB-Pyra Nbs in such ROS-high and -low tumors."

- The thioketal linkage has been intensively used in the drug delivery field for ROS-mediated cleavage and release of payload in ROS-high tumor models even without use of external stimuli. The authors examined whether MNB-Pyra Nbs could be released in a ROS-high tumor model, but they monitored it only for two hours, which is not sufficient enough to see the ROS-mediated drug release. It means the authors should repeat the same experiments and monitor the cleavage possibility over 24 hours.

2. My comment "3. In Figure 4b, the tumor accumulation of MNB-Pyra Nbs reached a maximum and gradually decreased with time after that peak time. It is likely that Nbs reached in the tumor would remain for long time after receptor-mediated endocytosis into cancer cells. If that is true, the accumulation of the Nb conjugates would increase with time. But the finding was opposite to the expectation. This finding should be explained and discussed."

- In EGFR expressing cancer cells, the authors showed MNB-Pyra Nbs could be internalized inside cells, which may be due to Nb-EGFR receptor mediated endocytosis. It is quite natural to think that once MNB-Pyra Nbs arrived in the tumor tissue, they should be taken up by EGFR-expressing cancer cells. It does not have any relevance with Fc-mediated endocytosis and recycling of an antibody. The authors' response to my comment is absolutely wrong. Anyway, then, what expected next is the accumulation of MNB-Pyra Nbs would increase with time and reach plateau and be maintained over more than 6-12 hours.

However, in reality, the accumulation of MNB-Pyra Nbs in the tumor reached max at 4 h post injection and then gradually decreased with time. This tendency is quite different from peptide or antibody-targeted drug delivery. The authors should respond to this comment well with reasonable explanation or evidence.

Reviewer #3 (Remarks to the Author):

I'm satisfied with the answers and corrections.

Reviewer #1 (Remarks to the Author):

The authors have addressed most of the points raised by this reviewer. Still, a few small adjustments for finalization are encouraged:

1. Nanobodies (Nbs), the smallest antigen-binding fragments with high stability and affinity derived from the variable domain of naturally occurring heavy-chain-only antibodies in camelids, bring a new opportunity to improve the specificity to tumors for photodynamic therapy (PDT) -> this is not the novel aspect of this article. So here should be correctly phased e.g. already shown to be an opportunity to improve... PDT.

Response: Thanks for your advice. We have revised this sentence in the manuscript.

The change in the manuscript:

Nanobodies (Nbs), the smallest antigen-binding fragments with high stability and affinity derived from the variable domain of naturally occurring heavy-chain-only antibodies in camelids, have been shown as an efficient way to improve the specificity to tumors for photodynamic therapy (PDT).

2. Moreover, during therapy, part of the long-term resident photosensitizers at the tumor site will be removed as the scab falls off as shown in the following picture. -> after illumination the PS is likely not functional anymore, as usually photobleaching indicates its exhaustion/ full use. The authors could add clarification on this aspect.

Response: Thanks for your advice. We further evaluated the photostability of the photosensitizer in vitro. As illustrated in Supplementary Figure 5a and b, the absorbance of the MNB-Pyra photosensitizer only exhibited a slight decrease (~10%) after the light irradiation (630 nm, 50 mW/cm²) for 60 min, which manifested its high photostability. Moreover, the efficient ROS generation of the photosensitizer was still observed by DHR123 probe after the irradiation (630 nm, 50 mW/cm², 60 min) (Supplementary Figure 5c). Therefore, this photosensitizer with high photostability is suitable for sustainable PDT because there is no need to worry about the exhaustion of the photosensitizers after multiple time of illumination.

The change in the manuscript:

The high photostability of MNB-Pyra was validated by the absorption with only a slight decrease after the light irradiation (630 nm, 50 mW/cm²) for 60 min, and then its ROS generation ability was also not affected (Supplementary Figure 5).

Supplementary Figure 5. (a) and (b) Photostability validation of **MNB-Pyra** (10 μ M) under light irradiation (630 nm, 50 mW/cm²) for 60 min. (c) ROS generation of **MNB-Pyra** (10 μ M) detected by DHR123 probe after the irradiation (630 nm, 50 mW/cm²) for 60 min.

3. The Nb format such as biparatopic may be an efficient way to prolong the circulation of Nbs in vivo -> that is not the case, there is no substantial prolongation of half-life with biparatopics, compared to monovalent. Later it is indicated the half-life of monovalent Nbs is of several hours but usually 60-90 min are referred in literature.

Response: Thanks for your correction. We have learned a lot from your suggestions. We also revised the related description in the manuscript.

The change in the manuscript:

Nevertheless, Nbs are rapidly removed from mammals via the circulation with an elimination half-life of 60-90 min due to their small size.^{17,28,29}

4. Sabrina Oliveira and her colleagues first conjugated Nbs with photosensitizers for nanobody-targeted PDT in 2014.²³ -> they first published on this approach in 2014.

Response: Thanks for your advice. We have revised this sentence in the manuscript.

The change in the manuscript:

Sabrina Oliveira and her colleagues first published on the approach of nanobody-targeted PDT in 2014,²³ and some other nanobody-targeted PDT related works were reported later.²⁴⁻²⁷

5. In these works, some progresses have been made but the photosensitizer used for conjugation is IRDye700DX only, which is a kind of conventional photosensitizers... -> this is a hydrophilic PS much different than most commonly used PS, so this sentence is misleading.

Response: Thanks for your correction. We have revised this sentence in the manuscript.

The change in the manuscript:

In these works, some progresses have been made but the photosensitizer used for conjugation is IRDye700DX only, while this hydrophilic photosensitizer still faces the limitation of tumor hypoxia environment during PDT.

Reviewer #2 (Remarks to the Author):

Although this revision seems to address some of my concerns well, there are still unclear responses to two of major comments raised previously. Thus, another revision is necessary.

1. My comment "Certain tumors produce high level of ROS in their microenvironment, for example, A549-derived tumor is one of them. It suggests that MNB-Pyra Nbs could release MNB-Pyra in such ROS-high tumors even without irradiation. This possibility should be tested by intratumoral injection of MNB-Pyra Nbs in such ROS-high and -low tumors."

- The thioketal linkage has been intensively used in the drug delivery field for ROS-mediated cleavage and release of payload in ROS-high tumor models even without use of external stimuli. The authors examined whether MNB-Pyra Nbs could be released in a ROS-high tumor model, but they monitored it only for two hours, which is not sufficient enough to see the ROS-mediated drug release. It means the authors should repeat the same experiments and monitor the cleavage possibility over 24 hours.

Response: Thanks for your advice. We reconstructed an A549 and an A431 tumor-bearing Nu/J mice models to investigate whether the photosensitizers could be released in tumor tissue over time. Notably, most of nanobody conjugates may be cleared before the thioketal linkage was cleaved due to the quick clearance of MNB-Pyra Nbs in vivo, which is not conducive to monitor the cleavage possibility over 24 hours. Therefore, the MNB-Pyra dimer with long-term retention was used for the intratumoral injection to observe the ROS-mediated drug release.

As shown in Supplementary Figure 26, the fluorescence signal at the tumor sites increased over time after extended the observation time, especially in A549 tumor model, indicating the cleavage of the connecting chain and release of photosensitizers. Therefore, the MNB-Pyra dimer could release monomer photosensitizers in ROS-high tumors without irradiation during the long time of observation. The fluorescence in A431 tumor model also exhibited increasement after 36 h, which is may ascribed to the much higher level of ROS in cancer cells than normal cells due to their increased metabolism and mitochondrial dysfunction.^{1,2}

Consequently, it's possible that MNB-Pyra Nbs could release monomer photosensitizers in ROS-high tumors even without irradiation. In our work, the monomer photosensitizers were released by illumination (630 nm, 30 mW/cm², 20 min) after the accumulation of MNB-Pyra Nbs at the tumor site (4 h in A431 tumor model), which is a rapid and concentrated release of monomers. But the effective response of the thioketal linkage to ROS in A431 cells was more than 12 hours. Thus, we believe the influence of this cleavage by intracellular ROS was negligible in this work.

References:

- [1]. Galadari, S., Rahman, A., Pallichankandy, S. & Thayyullathil, F. Reactive oxygen species and cancer paradox: To promote or to suppress? *Free Radical Biology and Medicine* **104**, 144-164 (2017).
- [2]. Jia, P. et al. The role of reactive oxygen species in tumor treatment. *RSC Advances* **10**, 7740-7750 (2020).

The change in the manuscript:

In order to confirm whether the level of ROS in cancer cells will cleave the thioketal linkage, MNB-Pyra dimer (0.3 mM, 100 μ L) was intratumorally injected in an A431 tumor-bearing mouse model for a long time of observation. As shown in Supplementary Figure 26, the fluorescence signal at the tumor sites increased over time in 36 h, indicating the cleavage of the connecting chain and release of photosensitizers. But the effective response of the thioketal linkage to ROS in A431 cells was more than 12 hours, which suggested the cleavage by intracellular ROS would have negligible influence in this work.

Supplementary Figure 26. Fluorescence observation after intratumorally injection of MNB-Pyra dimer (0.3 mM, 100 μ L) in A431 and A549 tumor models.

2. My comment "3. In Figure 4b, the tumor accumulation of MNB-Pyra Nbs reached a maximum and gradually decreased with time after that peak time. It is likely that Nbs reached in the tumor would remain for long time after receptor-mediated endocytosis into cancer cells. If that is true, the accumulation of the Nb conjugates would increase with time. But the finding was opposite to the expectation. This finding should be explained and discussed."

- In EGFR expressing cancer cells, the authors showed MNB-Pyra Nbs could be

internalized inside cells, which may be due to Nb-EGFR receptor mediated endocytosis. It is quite natural to think that once MNB-Pyra Nbs arrived in the tumor tissue, they should be taken up by EGFR-expressing cancer cells. It does not have any relevance with Fc-mediated endocytosis and recycling of an antibody. The authors' response to my comment is absolutely wrong. Anyway, then, what expected next is the accumulation of MNB-Pyra Nbs would increase with time and reach plateau and be maintained over more than 6-12 hours. However, in reality, the accumulation of MNB-Pyra Nbs in the tumor reached max at 4 h post injection and then gradually decreased with time. This tendency is quite different from peptide or antibody-targeted drug delivery. The authors should respond to this comment well with reasonable explanation or evidence.

Response: Thanks for your explanation. We apologize for not understanding your question correctly.

First, according to your question “It is likely that Nbs reached in the tumor would remain for long time after receptor-mediated endocytosis into cancer cells”, we investigated whether the MNB-Pyra Nbs would remain for long time after into cancer cells. As show in Response Figure 1, the uptake of MNB-Pyra Nbs increased over time and reached plateau in 90 min, and then the fluorescence signal in cells maintained for hours when the medium keeps unchanged. However, upon we removed the medium after the fluorescence signal reached plateau, the fluorescence in cells decreased obviously in 2 h indicating the drugs were excreted from the cells (Response Figure 2). These results confirmed that the drug concentration in surrounding environment would affect the retention of MNB-Pyra Nbs in cells.

As previously work reported,^{3,4,5} the half-life of Nbs in the bloodstream is significantly shorter (several hours, even 1.5 h) than full-length antibodies (21-days for IgG1), which indicated the drug concentration of MNB-Pyra Nbs in the blood may decreased to the half of its initial concentration after a few hours. The rapidly decreased drug concentration in the tumor environment is obviously not conducive to the retention of Nbs in cells. Therefore, the fluorescence signal would decrease after reached the maximum. Similar situations were also observed in other nanobody-related studies.^{6,7,8}

Second, our description of the way of Nbs enter the cells is not accurate. When Nbs were used alone, it was taken up by fluid phase uptake; only when Nbs were combined with other Nbs as biparatopic format or on nanoparticles, it led to EGFR clustering and is more efficiently internalized by receptor mediated endocytosis.^{9,10,11} Herein, we used Nbs alone, thus we have revised the description in the first revision as below but we never mentioned this in this question:

The nanobody 7D12 specifically binds to the ligand-binding domain III of the EGFR and is subsequently endocytosed into the cells by fluid phase uptake.

Cells continuously take up fluid from their neighborhood by a process designated as fluid-phase endocytosis, which is also called pinocytosis.^{12,13} This is different from the way of antibody enter cells through receptor-mediated endocytosis (Response Figure 3). Therefore, the tendency of MNB-Pyra Nbs is quite different from peptide or antibody-targeted drug delivery.

Response Figure 1. Uptake of MNB-Pyra Nbs (0.2 μ M) in A431 cells over time. Scale bar = 20 μ m.

Response Figure 2. Cells incubated with MNB-Pyra Nbs (0.2 μ M) for 2 h firstly, then the medium was replaced with new fresh medium and the fluorescence was observed (0-120 min). Scale bar = 20 μ m.

Response Figure 3. There are three types of endocytosis: phagocytosis, pinocytosis, and receptor-mediated endocytosis.

References:

[3]. Harmsen, M.M., Van Solt, C.B., Fijten, H.P. & Van Setten, M.C. Prolonged in vivo residence

- times of llama single-domain antibody fragments in pigs by binding to porcine immunoglobulins. *Vaccine* **23**, 4926-4934 (2005).
- [4]. Kijanka, M. et al. Rapid optical imaging of human breast tumour xenografts using anti-HER2 VHHs site-directly conjugated to IRDye 800CW for image-guided surgery. *European Journal of Nuclear Medicine and Molecular Imaging* **40**, 1718-1729 (2013).
- [5]. Mould, D.R., Sweeney, K.R. The pharmacokinetics and pharmacodynamics of monoclonal antibodies—mechanistic modeling applied to drug development. *Curr Opin Drug Discov Devel* **10**, 84–96 (2007).
- [6]. Bernhard, W., Barreto, K., El-Sayed, A., DeCoteau, J. & Geyer, C.R. Imaging Immune Cells Using Fc Domain Probes in Mouse Cancer Xenograft Models. *Cancers* **14**, 300 (2022).
- [7]. Peng, Q., Xiong, T., Ji, F., Ren, J. & Jia, L. Reduction-Activatable Fluorogenic Nanobody for Targeted and Low-Background Bioimaging. *Analytical Chemistry* **95**, 2804-2811 (2023).
- [8]. Xiao, Y. et al. Identification of a CEACAM5 targeted nanobody for positron emission tomography imaging and near-infrared fluorescence imaging of colorectal cancer. *European Journal of Nuclear Medicine and Molecular Imaging* **50**, 2305-2318 (2023).
- [9]. Schmitz, Karl R., Bagchi, A., Roovers, Rob C., van Bergen en Henegouwen, Paul M.P. & Ferguson, Kathryn M. Structural Evaluation of EGFR Inhibition Mechanisms for Nanobodies/VHH Domains. *Structure* **21**, 1214-1224 (2013).
- [10]. Roovers, R.C. et al. A biparatopic anti-EGFR nanobody efficiently inhibits solid tumour growth. *International Journal of Cancer* **129**, 2013-2024 (2011).
- [11]. Huang, C.Y. et al. A Novel Cellular Protein, VPEF, Facilitates Vaccinia Virus Penetration into HeLa Cells through Fluid Phase Endocytosis. *Journal of Virology* **82**, 7988-7999 (2008).
- [12]. Haigler, H.T., McKanna, J.A. & Cohen, S. Rapid stimulation of pinocytosis in human carcinoma cells A-431 by epidermal growth factor. *Journal of Cell Biology* **83**, 82-90 (1979).
- [13]. Lucero, D. et al. Interleukin 10 promotes macrophage uptake of HDL and LDL by stimulating fluid-phase endocytosis. *Biochimica et Biophysica Acta (BBA) - Molecular and Cell Biology of Lipids* **1865**, 158537 (2020).

REVIEWER COMMENTS

Reviewer #2:

This second round of revision have addressed the first main comment of mine, but failed to address the second main comment. Thus, I suggest another round of revision.

- The authors responded that internalization of nanobody alone occurs via pinocytosis. But most of targeting ligands that can interact with membrane bound receptors in cancer cells are taken up by receptor-mediated endocytosis, which is also demonstrated in the case of nanobodies (please do googling or literature survey, then the authors will find many examples of nanobody-receptor mediated endocytosis cases). Rather, it is unusual that receptor-bound nanobody is internalized via a merely cell's drinking water process, pinocytosis.

- In general, drug delivery system lacking a cancer-affinity ligand shows a maximum accumulation in the tumor at certain time and gradual decrease with time as a result of washing from the tumor. However, with high-affinity ligand, it would not be washed out from the tumor; instead, it reaches maximum and the plateau would maintain for hours. But, the pattern of tumor accumulation by MNB-Pyra Nbs, despite the presence of high-affinity Nbs, seems quite different from what is supposed to be observed with similar antibody-drug conjugates.

Reviewer #2 (Remarks to the Author):

This second round of revision have addressed the first main comment of mine, but failed to address the second main comment. Thus, I suggest another round of revision.

- The authors responded that internalization of nanobody alone occurs via pinocytosis. But most of targeting ligands that can interact with membrane bound receptors in cancer cells are taken up by receptor-mediated endocytosis, which is also demonstrated in the case of nanobodies (please do googling or literature survey, then the authors will find many examples of nanobody-receptor mediated endocytosis cases). Rather, it is unusual that receptor-bound nanobody is internalized via a merely cell's drinking water process, pinocytosis.

Response: Thanks for your advice. We modified the way of the conjugate enter cells in the last revision according to the suggestion of another reviewer: “The nanobody 7D12 specifically binds to the ligand-binding domain III of the EGFR and is subsequently endocytosed into the cells by fluid phase uptake”. After reconsideration, we found that the key of the endocytosis of 7D12 nanobody is the binding of nanobody with EGFR firstly, which is not consistent with the definition of pinocytosis that “this kind of endocytosis is not preceded by a specific binding to the specific sites on plasma membrane”. Meanwhile, the “fluid phase uptake” is a specific description of the process of nanobody entry into cells after the nanobody binds to EGFR. But we misunderstood the reviewer's suggestion and incorrectly attributed this way of entering cells to pinocytosis. The experiment results in our manuscript also demonstrated that the internalization of this antibody is significantly correlated with the EGFR expression of cells (Response Figure 1). Consequently, the way of this nanobody enter cells is still receptor-mediated endocytosis.

Response Figure 1: (a) Western blot analysis of the EGFR expression in A431, HeLa, A549, and NIH-3T3 cells. (b) Quantification of the fluorescence intensity of MNB-Pyra Nbs in different cells. Data are shown as mean \pm SD (n = 3).

The change in the manuscript:

The nanobody 7D12 specifically binds to the ligand-binding domain III of the EGFR and is subsequently endocytosed into the cells.

- In general, drug delivery system lacking a cancer-affinity ligand shows a maximum accumulation in the tumor at certain time and gradual decrease with time as a result of washing from the tumor. However, with high-affinity ligand, it would not be washed out from the tumor; instead, it reaches maximum and the plateau would maintain for hours. But, the pattern of tumor accumulation by MNB-Pyra Nbs, despite the presence of high-affinity Nbs, seems quite different from what is supposed to be observed with similar antibody-drug conjugates.

Response: Thanks for your comment. The Nbs conjugate in our work have a short residence time in vivo, which seems to be different from the behavior of traditional antibody conjugations such as monoclonal antibodies in vivo. In order to explain this difference, we first should understand the specific process of their entry into cells and seek answers from previous relevant works.

The process of receptor-mediated endocytosis is shown in Response Figure 2. The ligand arrives and attaches to receptors on the cell membrane, and then the membrane folds inward and forms vesicles, which later fuses with an endosome. After the early endosomes mature into late endosomes, the receptor and ligands were separated here. The receptor proteins then recycle back to the plasma membrane. [1,2] Therefore, the separation of ligand and receptor in the cells is necessary. The high affinity of antibodies indicates their rapid target recognition capacity and strong stability of the resultant antibody-antigen complex,[3] but it seems unlikely that the high affinity of the ligand can guarantee its long-term binding to the receptor in this process or prevent the ligand from being expelled from the cells. Meanwhile, after the separation of ligand and receptor, the free ligand should be able to be cleared from the cells over time, otherwise, the ligands can accumulate indefinitely in the cells. The monoclonal antibody-drug conjugates exhibit maintained plateau indicating its slow excretion rate after reaching maximum.

Response Figure 2: Specific process of receptor-mediated endocytosis.

The drug accumulation is a dynamic process, which means the uptake and efflux of drug occur simultaneously. The maintained plateau for hours in conventional antibody-drug conjugates indicates the close uptake and efflux rate in this period. For Nbs, they

enter the cells in the same way as full-length antibodies based on the receptor-mediated endocytosis process with similar uptake rate. However, the small size (~15kDa, compared with the 150kDa size of monoclonal antibodies) of Nbs exhibited faster clearance from the cells, [4,5] which is difficult to maintain the stable accumulation of drugs for hours in cells.

Moreover, the persistence of drug accumulation would be affected by the concentration of drugs in the bloodstream. Full-length monoclonal antibodies have the half-life over weeks, suggesting negligible change of drug concentration in the blood after few days.[6] But for Nbs, the half-life of Nbs in the bloodstream is significantly shorter (several hours, even 1.5 h) than monoclonal antibodies,[7] which is also obviously not conducive to the further accumulation of Nbs in cells and maintain a plateau.

Nbs are the smallest antigen-binding fragments with higher stability and affinity, due to their different properties from monoclonal antibodies, it is common to observe their imaging performance is inconsistent with that of monoclonal antibodies in vivo. But in essence, the accumulation pattern of MNB-Pyra Nbs in tumors is not fundamentally different from that of antibody-drug conjugates, because it is still a process of drug accumulation through receptor-mediated endocytosis and then the conjugate is metabolic cleared over time. It's just that these processes happen on a much smaller time scale. The absence of the signal plateau of Nbs was ascribed to their rapid clearance in cells and inadequate supply in the bloodstream after few hours.

Similar results can be found in previous Nbs-related imaging works. As shown in Response Figure 3, the rapid accumulation and clearance of the fluorescent nanobody-conjugate was observed in the tumor tissue unless the conjugate was further incorporated with the albumin binding domain (ABD).[8] The same tendency with our work was observed in the published work of Sophie Hernot, in which they connected the nanobody 2Rs15d with a fluorescent molecule IRDye680RD (Response Figure 4).[9] Besides, Sabrina Oliveira and her colleagues have compared the imaging performance of nanobody-conjugate 11A4-IR and monoclonal antibody-conjugate trastuzumab-IR in mice (Response Figure 5). As a result, the nanobody-conjugate led to a much faster tumor accumulation with high tumor to background ratios as compared to trastuzumab-IR due to its high affinity and quick clearance in blood (small size).[10]

Response Figure 3: Representative fluorescence images post-injection of Nb41-ABD-IR800 or Nb41-IR800 (n = 4).[8]

Response Figure 4: Representative fluorescence images of IRDye680RD labeled-nanobody 2Rs15d in mice.[9]

Response Figure 5: Comparison of optical imaging using labelled 11A4-IR (nanobody conjugate) and trastuzumab-IR (monoclonal antibody conjugate). [10]

References:

- [1]. Brown, V.I. & Greene, M.I. Molecular and Cellular Mechanisms of Receptor-Mediated Endocytosis. *DNA and Cell Biology* **10**, 399-409 (1991).
- [2]. Pathak, C. et al. Insights of Endocytosis Signaling in Health and Disease. *International Journal of Molecular Sciences*, **24**, 2971(2023).
- [3]. Chames, P., Van Regenmortel, M., Weiss, E. & Baty, D. Therapeutic antibodies: successes, limitations and hopes for the future. *Br. J. Pharm.* **157**, 220–233 (2009).
- [4]. Batra, S.K., Jain, M., Wittel, U.A., Chauhan, S.C. & Colcher, D. Pharmacokinetics and biodistribution of genetically engineered antibodies. *Current Opinion in Biotechnology* **13**, 603-608 (2002).
- [5]. Sapra, P. et al. Anti-CD74 Antibody-Doxorubicin Conjugate, IMMU-110, in a Human Multiple Myeloma Xenograft and in Monkeys. *Clinical Cancer Research* **11**, 5257-5264 (2005).
- [6]. Mould, D.R., Sweeney, KR. The pharmacokinetics and pharmacodynamics of monoclonal

- antibodies—mechanistic modeling applied to drug development. *Curr Opin Drug Discov Devel* **10**, 84–96 (2007).
- [7]. Lin, J.H., Guo, Y. & Wang, W. Challenges of Antibody Drug Conjugates in Cancer Therapy: Current Understanding of Mechanisms and Future Strategies. *Current Pharmacology Reports* **4**, 10-26 (2018).
- [8]. Xiao, Y. et al. Identification of a CEACAM5 targeted nanobody for positron emission tomography imaging and near-infrared fluorescence imaging of colorectal cancer. *European Journal of Nuclear Medicine and Molecular Imaging* **50**, 2305-2318 (2023).
- [9]. Debie, P. et al. Effect of Dye and Conjugation Chemistry on the Biodistribution Profile of Near-Infrared-Labeled Nanobodies as Tracers for Image-Guided Surgery. *Molecular Pharmaceutics* **14**, 1145-1153 (2017).
- [10]. Kijanka, M. et al. Rapid optical imaging of human breast tumour xenografts using anti-HER2 VHHs site-directly conjugated to IRDye 800CW for image-guided surgery. *European Journal of Nuclear Medicine and Molecular Imaging* **40**, 1718-1729 (2013).

REVIEWERS' COMMENTS

Reviewer #2 (Remarks to the Author):

Two main comments of mine are addressed properly and thus no further comments remain.